# Data Augmentations for Arithmetic Length Generalization in Small Transformers

## Abstract

Transformers often struggle with achieving length generalization on algorithmic tasks. To date, the most successful techniques attempting to achieve length generalization impose modifications to the model architecture. Instead, we propose Aligned Blankspace Augmentation (ABA), a simple data augmentation method that zero-pads numbers and inserts synchronized blank spaces across operands, demonstrating that the original Transformer architecture can achieve length generalization. Experiments demonstrate that small Transformers trained on up to 20-digit addition using our method achieve high accuracy on 200-digit problems, significantly outperforming prior works. The approach also enhances performance on other tasks like sorting and multi-operand addition, and improves multiplication generalization with scratchpads.

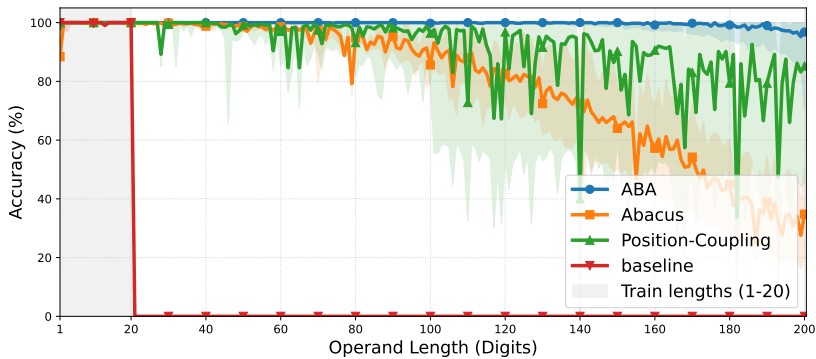

Figure 1: **Methods for length generalization on the addition task.** Accuracy is reported as the median over five runs; shaded regions show variation across seeds. ABA refers to our fixed-length variant Sec. 3.2, instantiated with a 6-layer decoder-only Transformer using relative positional embeddings (RPE). The baseline uses the same model and data without ABA. Position Coupling (Cho et al., 2024) and Abacus Embeddings (McLeish et al., 2024) are state-of-the-art methods for arithmetic length generalization. With ABA-fixed+RPE, a model trained on 1–20-digit addition generalizes robustly to 200-digit addition, attains the highest accuracy among all compared methods.

# 1 Introduction

Despite the success of Large Language Models (LLMs) (Brown et al., 2020; Chowdhery et al., 2022), recent studies have highlighted that Transformers (Vaswani et al., 2017), the backbone of LLMs, often cannot achieve length generalization on algorithmic tasks (Wu et al., 2024; Anil et al., 2022; Press et al., 2022). Length generalization tests whether a model can solve the same task at larger problem sizes than it saw in training. Complex scientific computations or multi-step reasoning often have unpredictable input lengths, making length generalization a critical test for precision and scale. The inability of the Transformer to length generalize also raises fundamental questions about how Transformers learn: for instance, do Transformers truly comprehend arithmetic processes, or do

they primarily rely on memorization from training? Thus, understanding the mechanisms of length generalization and developing methods to enhance this ability in Transformers is crucial.

In particular, Transformers struggle to achieve length generalization even for basic arithmetic tasks like addition (Kazemnejad et al., 2023; Nogueira et al., 2021; Lee et al., 2023). Zhou et al. (2023) identify index arithmetic—retrieving the matching digit—as a key bottleneck. To bypass this, they propose Index Hints, which place the same indexing symbols (e.g., 'a', 'b') before relevant digit pairs, so the model can easily perform indexing with induction heads instead of positional lookup. This clever method brought early success in length generalization (Zhou et al., 2024). Later, building on the core idea of explicitly tagging relevant digits, Cho et al. (2024) proposed Position Coupling and McLeish et al. (2024) introduced Abacus embeddings. Both methods work by assigning the same IDs into the positional encoding of relevant token pairs and deliver much stronger length generalization in arithmetic. More recently, Fan et al. (2025) showed that looped Transformers with an adaptive number of steps also improve length generalization. Thus, the current state of the art on arithmetic length generalization often relies on architecture modification.

This poses a key question: is architecture modification necessary to achieve robust length generalization? If the model can reach high in-distribution accuracy, extra machinery should not be necessary. We posit that the Transformer can extend the rule learned on short inputs to longer operands by padding operands while preserving the relative positional relationships between corresponding digits. To this end, we propose **Aligned Blankspace Augmentation (ABA)**, a data augmentation strategy that inserts blank spaces at identical relative indices within operands and results. For instance, the addition problem $a_1b_1c_1 + a_2b_2c_2 = a_3b_3c_3$ is transformed into $\square a_1 b_1 \square \square c_1 + \square a_2 b_2 \square \square c_2 = \square a_3 b_3 \square \square c_3$, where $\square$ represents a space token used for padding. It relies solely on the insertion of blank space tokens within the data, enabling Transformers to learn the task's intrinsic logic, rather than changing the Transformer's architecture or relying on an external scheme that labels matching digits. The effectiveness of this approach is clearly demonstrated by ABA's superior capability for length generalization: as illustrated in Fig. 1, it outperforms the state-of-the-art methods.

Our key contributions are threefold: Firstly, we empirically show that ABA enables long-range length generalization, outperforming prior methods. Secondly, we validate the broad applicability and robustness of ABA across various positional encodings and showcase its effectiveness on diverse algorithmic tasks, notably improving multi-digit multiplication when combined with a scratchpad. Thirdly, we provide a theoretical analysis, leveraging concepts from communication complexity and Limit Transformers (Huang et al., 2025) to offer insights into how ABA's preservation of alignment facilitates length generalization, particularly in scenarios where other data formats struggle.

## 2 RELATED WORK

**Positional Encoding and Data Format Engineering** Positional encoding (PE) critically influences length generalization. FIRE, No Positional Encoding (NoPE), and various relative schemes have shown promise (Kazemnejad et al., 2023; Zhou et al., 2024). Another line of research focuses on data formatting; Lee et al. (2023) show that reversing the output makes addition a local, position-wise mapping and leads to improvement in both performance and sample efficiency. They also introduced a zero-padded format for multi-digit addition, where operands and outputs are padded to a fixed width to give the transformer a fixed-length representation for each example. Shen et al. (2023) propose Random Spacing, achieving near-perfect generalization from 10-digit to 12-digit addition. More recently, Sabbaghi et al. (2024) show that by right-aligning numbers and left-padding to a uniform length, encoder-only Transformers with RPE achieve strong length generalization.

**Position Markers** Position markers aim to explicitly highlight digit significance for better alignment. Index Hints (Zhou et al., 2023) achieve this by prepending symbolic markers to relevant digits, achieving up to $2.5\times$ generalization in addition. Other approaches like Position Coupling (Cho et al., 2024) and Abacus embeddings (McLeish et al., 2024) assign consistent positional encodings to digits of identical significance, increase the ratio of generalization to $6\times$.

**Theory of Length Generalization in Transformers** RASP-L (Zhou et al., 2023) offers a theoretical lens on what algorithms Transformers can learn, suggesting that if a task has a short RASP-L program that generalizes across lengths, the model tends to acquire a single position-uniform procedure reused

at all positions. This view has been widely recognized, and recent work by Huang et al. (2025) extends it with a formal framework.

Due to space constraints, we provide an extended related works section in Appx. A.

## 3 ALIGNED BLANKSPACE AUGMENTATION

### 3.1 THE CHALLENGE OF DIGIT ALIGNMENT IN ARITHMETIC GENERALIZATION

A primary challenge for Transformers in arithmetic length generalization is identifying corresponding digits based on their significance across varying sequence lengths, especially in decimal addition. As highlighted by Zhou et al. (2023), successfully identifying these digits requires index-related operations. Without explicit position markers, the model tends to perform digit identification by absolute position look-up (e.g. reading the token at position 7) rather than position-uniform index arithmetic (e.g. finding the middle of the prompt sequence using relations that do not depend on sequence length). This strategy is tied to the specific indices seen during training and therefore fails to transfer to unseen positions at longer lengths.

On the other hand, there is evidence that the model reuses the same computation across different positions. Lee et al. (2023) demonstrate that with reverse (least-significant-bit-first) output, the next token at position $i$ ($y_i$) depends only on three inputs: the $i$-th digit of operand 1 ($a_i$), the $i$-th digit of operand 2 ($b_i$), and the carry from position $i - 1$ ($c_{i-1}$). Concretely,

$$y_i = (a_i + b_i + c_{i-1}) \bmod 10, \qquad c_i = \left\lfloor \frac{a_i + b_i + c_{i-1}}{10} \right\rfloor. \tag{1}$$

They also report that holding out the digit "5" from appearing as the most significant digit during training still yields high test accuracy when "5" appears at test time; corrupting earlier outputs does not prevent the model from predicting the next digit correctly. These findings suggest that the model learns a local rule for emitting each digit, but that this rule fails to reach further positions because of the indexing bottleneck. Specifically, the model may be unable to locate the correct digits $a_i$ and $b_i$ at unseen positions beyond its trained range. This motivates a simple intervention to length generalization: if we pad operands to longer lengths, can the Transformers learn to use the same per-digit formula at further positions?

One simple idea is to insert random blank spaces into the operands (Shen et al., 2023), which break consecutive digits and force them to appear at further absolute positions during training and, in principle, let the model practice producing digits at larger indices. However, this approach only yields marginal improvements in length generalization, extending by two digits when combined with a scratchpad. Our interpretation is that random spacing disrupts the consistent relative alignment between corresponding digits of the operands and the sum, which aggravates the indexing bottleneck and makes it even harder to locate correct digit pairs. We provide a theoretical explanation in Sec. 6 for why random spacing does not enable robust length generalization.

### 3.2 PRESERVING ALIGNMENT WITH ALIGNED BLANKSPACE AUGMENTATION

Based on the limitations above, we hypothesize that preserving the relative positional relationships between corresponding digits across the operands and the sum is crucial for length generalization in arithmetic tasks. We therefore introduce Aligned Blankspace Augmentation (ABA), which pads the data with blank spaces while simultaneously maintaining consistent alignment structures. We achieve this by:

1. **Zero-Padding:** First, each numerical component (the operands and the answer) in the equation is zero-padded to length $L$. This length $L$ is determined to be sufficiently long to accommodate the longest operand and the expected maximum length of the sum. For example, in an addition task, if $L_{o1}$ and $L_{o2}$ represent the original two operands lengths, $L$ is set to $\max\{L_{o1}, L_{o2}\} + 1$ to account for a potential carry digit in the sum. See Tab. 3 for the task-specific rules to determine $L$.

2. **Synchronized Space Insertion:** After the initial zero-padding, an identical pattern of blank spaces is applied to all numerical components to expand them to a target length, $p$. Concretely,

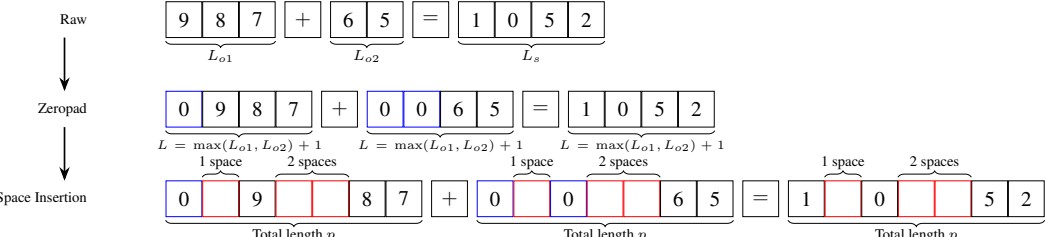

Figure 2: **Illustration of the Aligned Blankspace Augmentation (ABA) method on addition** $987 + 65 = 1052$ (numbers in standard order). The process includes: (1) The Raw input, where the first operand 987 has length $L_{o1} = 3$, and second operand 65 has length $L_{o2} = 2$; (2) Initial alignment via Zero-padding (blue boxes) to length $L = \max(L_{o1}, L_{o2}) + 1 = 4$ in the Zeropad stage; and (3) Synchronized Space Insertion (red boxes represent inserted blank spaces), where two blank spaces are inserted in the two operands and answers between their 2nd and 3rd significant digits (after zero-padding operands), and one space inserted between their 1st and 2nd significant digits, so that they achieve a uniform total length $p = 7$, while maintaining relative digit positions.

to determine the positions for the blank spaces in the numerical components, we first sample $L$ unique positions from the $p$ available slots uniformly at random without replacement. To preserve the original digit order, the $L$ randomly sampled positions are then sorted in ascending order. The digits of each numerical component are then inserted sequentially into this sorted layout, with any unfilled positions becoming blank spaces.

Figure 2 illustrates this ABA process. This aligned insertion ensures that each digit or space in the operand and sum maintains a consistent relative positioning, regardless of the absolute positions they occupy due to the inserted spaces. We propose two variants of ABA, with full algorithms in Appx. B.1 and examples in Appx. B.3. Below, we detail the differences between the two variants:

- **Variant 1 (ABA-fixed):**
  *Train:* The target length $p$ in every example is a fixed hyperparameter, $p_{\max}$.
  *Test:* Blank spaces are added to the right of every operand so operand length equals $p_{\max}$.

- **Variant 2 (ABA-var):**
  *Train:* The target length $p$ for each example is chosen uniformly at random $L$ up to $p_{\max}$.
  *Test:* No spaces are added to the input. Operands are zero-padded.

ABA-fixed guarantees strict alignment of corresponding digits across operands and sum, while ABA-var preserves weaker alignment but at test time we can feed space-free, zero-padded input, making the method usable in standard inference settings. To our knowledge, there is currently no other method that achieves strong length generalization on standard inputs without architectural changes, which gives ABA-var clear practical value. We wrap each sample with $ delimiters as both a start of sequence and end-of-sequence (EOS) marker. At test time, we take the model's output up to (but not including) the first $, then strip any generated blanks before comparing it to the ground truth.

## 4  EXPERIMENTS

**Setup:** Following the methodology of Lee et al. (2023), we focus on teaching arithmetic to small decoder-only Transformer models. We evaluate models using learned positional embeddings (Vaswani et al., 2017) as well as relative positional embeddings (RPE) (Shaw et al., 2018), inspired by Sabbaghi et al. (2024) who successfully length generalized using padding and RPE with an encoder-decoder model. Results with other positional embeddings on addition are shown in Sec. 4.2. We report results using exact-match accuracy, where each result is the median over five runs. Complete model configurations, training details, and datasets are summarized in Appx. C. We select Position Coupling and Abacus embeddings as our primary baselines, since they have demonstrated the strongest length generalization capabilities so far.

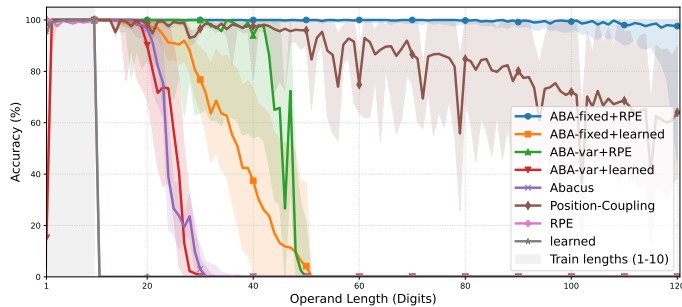

Figure 3: **Addition length generalization with Transformers under ABA-fixed and ABA-var.** Each variant is paired with learned or relative positional embeddings (RPE). ABA consistently outperforms non-ABA baselines across tested lengths. Curves show accuracy versus test operand length (operands have equal length). Training uses 1–10 digits; $p_{\max} = 121$ for ABA-fixed+RPE and $p_{\max} = 51$ for the other setups.

### 4.1 ABA ENHANCES LENGTH GENERALIZATION IN ADDITION

We begin with the task of addition. Following Lee et al. (2023), we adopt the reverse format (e.g. $31 + 425 = 735$) to reduce the task to a per-position mapping as demonstrated with Eq. (1). We evaluated our two primary ABA strategies, **ABA-fixed** (Variant 1) and **ABA-var** (Variant 2), each combined with Learned Positional Embeddings (learned) and Relative Positional Embeddings (RPE). The maximum target total length $p_{\max}$ for each numerical component (after zero-padding and synchronized space insertion, as defined in our ABA variants) was specifically configured for these setups. For the ABA-fixed with RPE (ABA-fixed+RPE) configuration, we set $p_{\max} = 121$. The three other setups all used $p_{\max} = 51$. We note that an excessively large $p_{\max}$ can impact training stability. Further ablation studies investigating the impact of model depth (number of layers), the range of operand digit lengths used during training, and the ABA parameter $p_{\max}$ are presented in Appx. E.

Our results, presented in Fig. 3, show that our ABA methods lead to significant improvements in length generalization compared to baseline Transformers without ABA. We observe that models employing the ABA-var strategy generally show less effective length generalization compared to ABA-fixed. This difference is understandable; our goal with ABA is to ensure alignment for corresponding digits. The ABA-fixed strategy, by enforcing a consistent total length for all components, provides a more uniform structure. While the ABA-var setup offers flexibility, its varying total lengths during training and testing may make it harder to achieve precise digit alignment. Importantly, for models trained with ABA, length generalization ceases as the operand length approaches $p_{\max} - 1$. This pattern suggests that the models learn to do arithmetic on the digit positions exposed by padding, but do not extend beyond them. A controlled comparison of other padding strategies (Shen et al., 2023; Sabbaghi et al., 2024) appears in Appx. D.1. Comprehensive ablations on model depth, $p_{\max}$, training length, and alignment are reported in Appx. E. We also report iteration time and memory in Tab. 8.

### 4.2 GENERALIZATION IN ADDITION UNDER DIVERSE CONDITIONS AND SETTINGS

**Generalization from 10+10**: Most studies on arithmetic length generalization train on operands of varying lengths. We thus investigated whether training on operands of varying lengths is necessary for length generalization. To test this, we constructed a training dataset consisting only of 10-digit + 10-digit addition problems. We then evaluated the model's generalization on the test set containing operand length pairs from 1 to 20 digits. For these experiments, we set $p_{\max} = 21$, matching the maximum operand length after zero-padding in the test set. As shown in Figure 4 (left), models trained with ABA-fixed successfully learned to perform addition for all operand lengths from 1 to 20 digits. In contrast, models with Position Coupling failed to achieve perfect generalization across all lengths. Results with Abacus embeddings and other ABA configurations and a sequence-length-matched evaluation of Position Coupling which pads all operands to 10 digits at test time are provided in Appx. D.2.1.

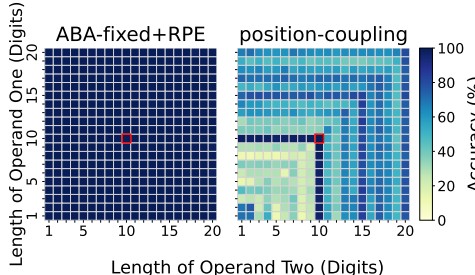 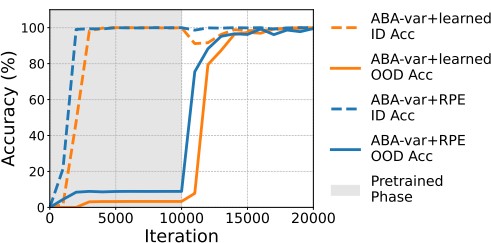

Figure 4: **Left: Accuracy comparison of ABA-fixed+RPE versus Position Coupling when models are trained *exclusively* on 10+10 digit addition.** The region enclosed by the red box represents the training set. Performance is evaluated on test operand pairs with lengths varying from 1 to 20 digits. **Right: Effect of applying ABA-var as a finetuning strategy.** A baseline model, initially pretrained on 1-10 digit addition (standard format), is finetuned using the ABA-var format on the same dataset. The plot displays the resulting in-distribution (ID, 1-10 digits) and out-of-distribution (OOD, 10-20 digits) accuracy throughout the training and finetuning iterations.

**Finetuning**: Given that ABA-var employs a standard, space-free format at test time, we investigated its efficacy for finetuning. Models pre-trained on the baseline format were subsequently finetuned on their original 1-10 digit dataset, now reformatted using ABA-var. Here, we again set $p_{\max} = 21$. As depicted in Fig. 4 (right), this finetuning process enhanced out-of-distribution accuracy for both models. This shows that ABA-var helps models carry the rules learned in pretraining to farther positions, highlighting its potential to improve length generalization in models initially trained on simpler formats. Further comparisons between ABA-var finetuning and training from scratch, including full learning curves, are shown in Appx. D.2.2.

**Other positional embeddings**: We examined whether ABA benefits models employing other types of positional embeddings commonly used in length generalization research. We tested several popular positional encoding schemes, including learned, RPE, NoPE, ALiBi (Press et al., 2022), Rotary (Su et al., 2023), FIRE (Li et al., 2024), Kerple (Chi et al., 2022), Sinusoidal (Vaswani et al., 2017). All experiments in this section use $p_{\max} = 21$. As illustrated in Tab. 1, performance varies considerably across different positional embedding methods, but applying ABA generally enhanced length generalization capabilities for most of the tested embeddings. Detailed per-length curves for these positional embeddings with and without ABA are in Appx. D.2.3.

Table 1: **Accuracy (%) of different positional embeddings with ABA strategies on addition.** Training uses operands with 1–10 digits, and testing uses operands with 1–20 digits.

| PE Type | without ABA | ABA-fix | ABA-var |
|---|---|---|---|
| NoPE | 43.16 | 30.25 | 55.97 |
| sinusoidal | 25.01 | 91.01 | **99.32** |
| learned | 25.00 | **100.00** | 99.80 |
| ALiBi | 34.44 | **91.05** | 75.92 |
| FIRE | 35.39 | 50.56 | 55.55 |
| Kerple | 39.39 | **95.32** | 56.03 |
| RoPE | 39.45 | 34.63 | 58.81 |
| RPE | 25.88 | **100.00** | **100.00** |

### 4.3 EXPERIMENTS ON ADDITIONAL ALGORITHMIC TASKS

In addition to standard two-operand addition, we evaluate the effectiveness of our proposed ABA strategy on a range of other algorithmic tasks known to be challenging for length generalization (Zhou et al., 2023; Jelassi et al., 2023; McLeish et al., 2024; Cho et al., 2024). These tasks include Copy with Repeated Tokens, Reverse with Repeated Tokens, $N$-digit × 2-digit Multiplication ($N \times 2$), Sorting, and Multi-Operand Addition (MultiAdd). Notably, for the Sort and MultiAdd tasks, to enable the model to generalize to a variable number of operands, in addition to inserting spaces within the digits of actual operands, we also introduce a random number of operand placeholders. These placeholders consist entirely of blank spaces and are padded to the same total length $p$ as the augmented numerical operands. They move operand positions farther to the right in training, enabling the model to learn to handle later-position operands, which supports generalization to more operands at test time. Visual examples of these task formats and the application of ABA are illustrated in Fig. 5.

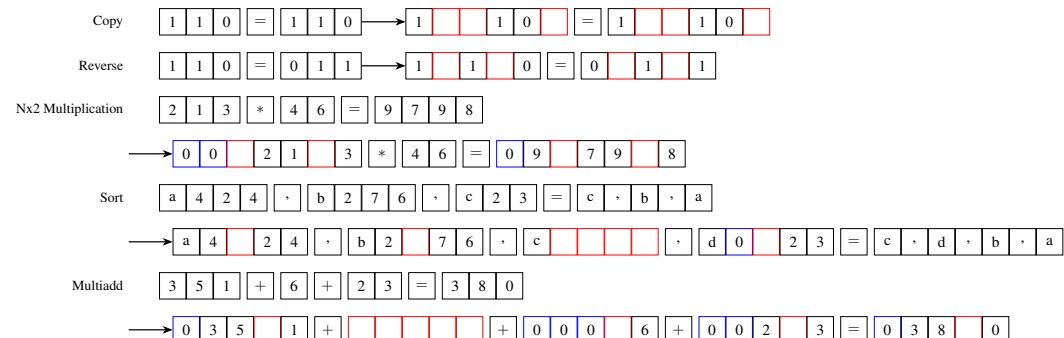

Figure 5: **Illustration of ABA applied to various algorithmic tasks.** For each task, the standard input format (top or left) is contrasted with its ABA-augmented version (bottom or right).

As shown in Tab. 2, ABA significantly aids the model in achieving length generalization across the other evaluated tasks. In our results, we use the following notation: in $X \rightarrow Y$, $X$ is the maximum training length and $Y$ is the maximum test length. In $a/b \rightarrow c/d$, $a$ is digits per operand and $b$ is the number of operands in training; $c$ and $d$ are the corresponding test settings. Both test and training dataset sweep from 1 up to the maximum length mentioned above. We set $p_{\max}$ equal to the longest operand digit length after zero-padding in the test set. Task definitions, data sampling method, and detailed results are in Appx. D.3. Across tasks, ABA generally matches or exceeds Position Coupling and Abacus embedding. This suggests that our proposed augmentation provides a broadly applicable mechanism for improving digit alignment and sequence processing, contributing to better length generalization capabilities in Transformers across diverse algorithmic challenges. Results on symbolic string tasks (Caesar cipher and string reversal) are included in Appx. D.3.3, demonstrating that ABA extends to symbolic transformations beyond arithmetic with numbers.

Table 2: **Performance comparison on various algorithmic tasks.** Column headers are abbreviated for brevity: LPE (Learned Positional Embeddings), RPE (Relative Positional Embeddings), ABA-F (ABA-fixed), ABA-V (ABA-var), Abacus (Abacus Embeddings), PC (Position Coupling).

| Task | LPE | RPE | LPE+ABA-F | LPE+ABA-V | RPE+ABA-F | RPE+ABA-V | Abacus | PC |
|---|---|---|---|---|---|---|---|---|
| copy:10→100 | 10.0 | 10.0 | 99.8 | 27.3 | 99.9 | **100.0** | 47.9 | 79.6 |
| reverse:10→100 | 10.1 | 10.1 | 67.8 | 22.1 | 45.5 | 87.3 | 13.6 | **100.0** |
| multi_add:5/5→10/10 | 4.4 | 4.8 | 62.9 | 5.0 | **69.2** | 5.1 | 11.8 | 33.5 |
| sort:5/5→10/10 | 6.8 | 14.6 | 51.2 | 49.6 | 39.0 | **55.6** | 23.3 | 38.6 |
| multiplication $N \times 2$:10→60 | 17.1 | 17.1 | 49.8 | 48.1 | 69.2 | 82.4 | 19.1 | **96.3** |

## 4.4 CARRY-CHAIN STRESS TESTS

A *carry chain* in arithmetic is the number of consecutive digit positions through which a carry must propagate. For example, in $999999 + 1$, the carry propagates through all six positions, so the carry chain length is six. To test whether a model *faithfully* length-generalizes on arithmetic, we would like to know if it can (i) handle carries that span the entire number, (ii) handle carries that grow with the number of operands. To this end, we run the following experiments:

- **Full-carry addition.** We train on integer addition where operand lengths and digits are sampled uniformly. For evaluation, we construct a synthetic test set of full-carry cases where the carry chain length equals the longer operand length (for example, $999 + 1 = 1000$).

- **Full-carry multi-operand addition.** We train on multi-operand addition with up to 5 operands and up to 5 digits per operand. For testing, we generate sums with $K \in \{1, \ldots, 10\}$ operands. Each operand length $\ell$ is sampled uniformly from $\{1, \ldots, 5\}$, and every digit is sampled independently from $\{7, 8, 9\}$. This construction forces large carries in every column, and carry chain has to propagate across all operands in the sum.

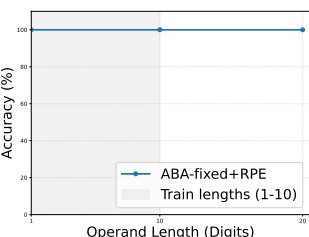 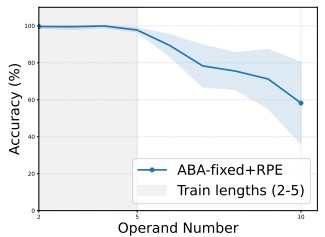 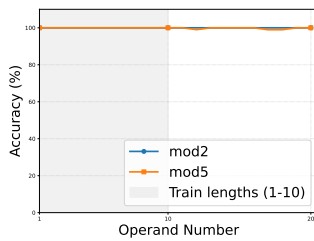

Figure 6: **Left:** Full-carry addition accuracy where the carry chain spans all digit positions, showing generalization from length-10 to length-20. **Middle:** High-carry multi-operand addition with $1$–$10$ operands, each of length $1$–$5$ digits, where all digits are drawn from $\{7, 8, 9\}$ to induce full carries. The experiment shows generalization from 5 operands to 10 operands. **Right:** Prefix-parity with scratchpad for $m = 2$ and $m = 5$, showing generalization from length-10 to length-20.

- **Parity with scratchpad.** We consider prefix-parity tasks that require a cumulative update across the entire prefix. Given an input sequence $(x_t)$, the target scratchpad sequence is $y_t = \left(\sum_{i=1}^{t} x_i\right) \bmod m$. We study $m = 2$ with $x_t \in \{0, 1\}$, and $m = 5$ with $x_t \in \{0, \dots, 4\}$. This task naturally requires the model to maintain and update a global summary over all previous positions. We show training on length of 1-50 and generalize to 200 in Fig. 20.

Across these three settings, the model must propagate information over long chains. We run these experiments with ABA-fixed+RPE, and the results are in Fig. 6. In all cases, the model maintains stable performance, which suggests that it has learned a digit-wise update rule which can extend to longer dependency lengths than those that appear in training examples.

## 5 ABA Scratchpad for Multiplication

While ABA demonstrated broad effectiveness across several algorithmic tasks, its direct application to multi-digit multiplication yielded limited length generalization (as shown in Fig. 7, left). This aligns with the recognized difficulty of multiplication for sequence models, often necessitating explicit intermediate reasoning steps (Nye et al., 2021). For instance, Shen et al. (2023) utilized a scratchpad based on summing M-digit × 1-digit multiplications, achieving high accuracy on in-distribution data. More recently, Cho et al. (2025) demonstrated length generalization to some extent by combining a multiply-then-add scratchpad with 3D Position Coupling.

Motivated by the potential to leverage ABA's alignment preservation capabilities within such a framework, we designed a scratchpad format for multiplication that integrates our ABA strategy. The core idea behind our scratchpad is to decompose the N×M multiplication into a series of addition problems. We iterate through the tokens (digits and spaces) of the spaced first operand from the least significant bit, where each digit initiates a calculation step of a N×1

```
Detailed Scratchpad

37 5 * 00 06 33  :
5* 00 06 33  = 00 31 65  > 00 31 65  + 00 00 00  = 00 31 65  ,
 * 00 06 33  =           >           + 00 31 65  = 00 31 65  ,
7* 00 06 33  = 00 44 31  > 04 43 10  + 00 31 65  = 04 74 75  ,
3* 00 06 33  = 00 18 99  > 18 99 00  + 04 74 75  = 23 73 75  ,
 * 00 06 33  =           >           + 23 73 75  = 23 73 75

Simplified Scratchpad

37 5 * 00 06 33  :5- 00 31 65  , -            ,7- 04 43 10  ,
3- 18 99 00  , -            ,
> 00 31 65  +          + 04 43 10  + 18 99 00  +           = 23 73 75
```

multiplication, shifting the result based on its significance, accumulating into a running sum, while each space generates a corresponding placeholder row. Meanwhile, ABA-fixed is applied to the second operand and all intermediate numerical values. They are then formatted to a final, fixed token length by inserting spaces at the *same relative indices* derived from the spacing pattern used in the first operand. The scratchpad generation protocol and concrete formatted examples are detailed in Appx. F.

We also designed a *simplified variant* which omits the running accumulation steps. It directly presents the correctly shifted and aligned partial products, framing the final calculation as a multi-operand

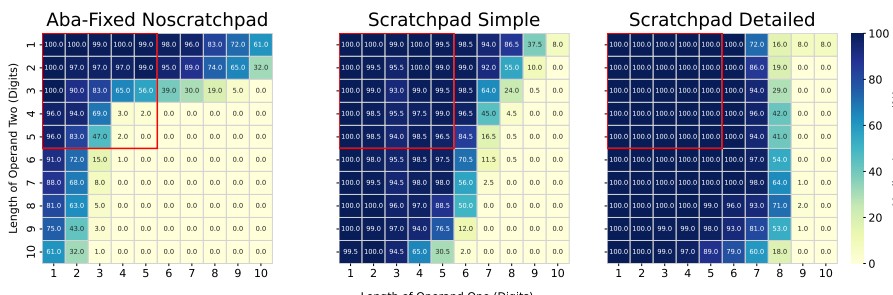

Figure 7: **Performance comparison on multiplication.** Models were trained on a 5×5 dataset and tested on operand length pairs ranging from 1 to 10 digits. Each cell's accuracy is the average over 100 test samples. The region enclosed by the red box represents the training set. For the scratchpad method, the first operand is padded to a variable length up to 10 tokens, while the second operand and intermediate representations are padded to $p_{max} = 21$ using ABA-fixed. For all the methods, at test time, the model receives only the problem statement (e.g., `234*234=`) as input, and accuracy is calculated based on the exact match of the final answer.

addition task. Full details of the scratchpad generation are provided in Appx. F. The above scratchpad illustrates an example of $375 \times 633$. For simplicity, all numbers in the scratchpad (i.e., intermediate steps) are in plain format instead of in reverse.

We show our result in Fig. 7. Generalization improves significantly with the scratchpad, and thus we show that our data format, when integrated into a structured computational trace, can effectively guide Transformers to achieve length generalization on complex algorithmic tasks like multiplication. Notably, while length generalization with respect to the length of the second operand is significantly improved by our ABA scratchpad, generalization with respect to the first operand is comparatively limited. The disparity is attributed to the scratchpad's mechanism, which effectively transforms the N×M multiplication into a multi-operand addition where the number of digits in the first operand dictates the number of terms to be summed. This observation is consistent with Cho et al. (2025) that generalizing over numbers of operands is inherently more challenging than generalizing over the length of operands.

## 6 Why Use Aligned Blank Spaces?

**Notation**: For a function $f$, we denote $f \in \Omega(g)$ to represent that $\exists n \in \mathbb{N}, c \in \mathbb{R}^+$ s.t. for $n \geq N$, $f(n) \geq cg(n)$. Similarly, $f \in \mathcal{O}(g)$ means that $\exists n \in \mathbb{N}, c \in \mathbb{R}^+$ s.t. for $n \geq N$, $f(n) \leq cg(n)$.

An initial hypothesis for why our method works is that it increases the computational power the transformer has access to in the operands. However, adding random blank spaces without any alignment between the operands also achieves this (Shen et al., 2023). Yet, empirically, this method does not work as well as ABA, as demonstrated in Appx. D.1. To understand why ABA outperforms this baseline, we use tools from communication complexity to mathematically formalize why alignment of blank spaces is necessary for length generalization.

In particular, Huang et al. (2025) formally demonstrate that there exists a particular transformer which length generalizes for a task, called a "limit transformer", only if the task has communication complexity at most $\mathcal{O}(\log N)$, where $N$ is the input length. We provide a brief background on communication complexity and limit Transformers in Appx. G. Addition of $K$-length operands with no augmentations has a communication complexity of at least $\Omega(K)$, providing theoretical justification for why Transformers fail to length generalize for standard addition. In particular, inserting random blank spaces into the operands up to input length $p_{\max}$ does not reduce this communication complexity, thus failing to satisfy this necessary condition:

**Proposition 6.1** (Informal). *$K$-length addition augmented with random blank spaces up to length $p_{\max}$ has a communication complexity of $\Omega(p_{\max})$, when $p_{\max} \geq K$, and thus a length generalizing limit transformer does not exist for this task.*

*Proof Sketch*: This follows by a reduction from the index problem, which also has linear communication complexity (Roughgarden, 2015). For a full proof, see Appx. H.1.

But, when addition is augmented with *aligned* blank spaces, like in our ABA algorithm, the $\mathcal{O}(\log N)$ communication complexity condition *is* satisfied for sufficiently many spaces:

**Proposition 6.2** (Informal). *$K$-length addition augmented with aligned blank spaces up to length $p_{\max}$ has a communication complexity of $\mathcal{O}(\log p_{\max})$, for sufficiently many spaces. Thus, a length generalizing limit transformer can exist for this task.*

*Proof Sketch*: Since Alg. 1 places spaces in the same positions for both operands, Bob already knows the space positions in Alice's operand. Then, Alice need only communicate the $\mathcal{O}(K)$ digits. As such, considering $p_{\max} \geq 2^K$, since $\mathcal{O}(K) = \mathcal{O}(\log p_{\max})$, we are done. For a full proof, see Appx. H.2.

**Discussion**: Firstly, satisfying the communication complexity condition of $\mathcal{O}(\log N)$ is necessary but not sufficient for a limit transformer to exist. Furthermore, a limit transformer is one such length generalizing transformer. However, as empirically demonstrated by Huang et al. (2025), the existence of a limit transformer highly correlates with success on length generalization. Secondly, as demonstrated in Prop. H.4, even when the same amount of spaces as Prop. 6.2 are added using random blank space augmentation, it obtains a communication complexity of $\Omega(\log^2 p_{\max})$ and thus still does not admit a limit transformer. Thirdly, the above results hold similarly for any task which has an $\Omega(N)$ communication complexity without any data augmentations, like copying a string. Finally, even when alignment is decreased slightly by e.g. adding random spaces after ABA, communication complexity is increased and length generalization suffers, as demonstrated in Appx. E.4.

## 7 DISCUSSION AND FUTURE WORK

The success of ABA shows a practical route to arithmetic length generalization without changing the architecture: pad operands to a longer length while keeping corresponding digits aligned. This encourages the model to apply the same per-digit operation at each position up to the maximum padded length $p_{\max}$. When queried with longer operands, the model keeps reapplying this rule across positions within $[1, p_{\max}]$ to produce longer outputs. This view lines up with the RASP-L conjecture that Transformers tend to learn a single per-digit, position-uniform procedure reused at all positions in length-generalizable tasks (Zhou et al., 2023). In effect, ABA shows that the model can even length generalize by learning each digit position separately.

However, for arithmetic tasks that require index arithmetic (e.g. addition), ABA alone does not provide the extra indexing signals, which would let a model use induction heads to compute solutions. As a result, the model falls back on cues from the positional encoding. This is not a limitation of our setting: by design, ABA is not meant to generalize beyond $p_{\max}$. Its role is to transfer the same digit-wise rule to all positions exposed by padding and achieve length generalization up to $p_{\max}$.

We also note that, while the term length generalization is often used in a broader sense to refer to generalization to longer sequence lengths, in this work we state explicitly that ABA-fixed is used as a method for operand-length generalization: it helps the Transformer solve problems with longer operands than those seen during training.

On an intuitive level, ABA shows that within a fixed aligned grid, a Transformer can at inference time execute more steps than are typical in training. This implies Transformers can scale up the amount of computation at inference by reusing a learned local rule on a larger set of positions, without adjusting the computational budget per step.

**Limitations and Future Work**: The insertion of spaces increases sequence length, leading to greater computational demands, which is a clear drawback compared with strong baselines such as Position Coupling and Abacus Embeddings that do not increase the training sequence length. In addition, its current formulation is most directly applicable to tasks with clear alignment structures; extending it to less uniformly structured tasks requires non-trivial adaptations. However, we show in Appx. E.4 how this can be partially relaxed. Future research will focus on several key directions. A primary objective will be the optimization of ABA's efficiency and practicality to reduce computational overhead and ensure robust transfer to standard, space-free inference settings, thereby enhancing practical deployment. Furthermore, developing principled approaches for hyperparameter determination $p_{\max}$ is essential.

## 8 REPRODUCIBILITY STATEMENT

We have taken several measures to ensure the reproducibility of our results. Our code-base, including dataset, model training script, and evaluation routines, is released at https://anonymous.4open.science/r/ABA-C4C1. Detailed hyperparameters for all models are provided in Appendix C.1 (Tables 4–7). Training procedures, including loss computation, checkpointing schedule, and number of iterations, are described in Appendix C.2. Dataset construction, and preprocessing rules for both addition and other algorithmic tasks are documented in Appendix C.3. Computational costs in terms of training time, memory usage, and inference efficiency are analyzed in Appendix C.4 (Table 8). Together, these resources should allow independent researchers to reproduce both our results.

## 9 ETHICS STATEMENT

This work focuses on improving the length generalization ability of Transformer models on synthetic arithmetic. Our study does not involve human subjects, sensitive personal data, or proprietary datasets. All experiments are conducted on synthetic data that we generate ourselves, and thus raise no concerns regarding privacy, safety, or fairness. The research does not present foreseeable risks of misuse or harmful applications. We have carefully followed the ICLR Code of Ethics and are committed to ensuring the integrity, transparency, and reproducibility of our work.

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

# Appendix

# Table of Contents

## A    EXTENDED RELATED WORKS

**Positional encodings and length generalization.** A large body of work shows that the choice of positional encoding (PE) is a major driver of length generalization. Systematic comparisons across absolute and relative schemes (Absolute Position Embeddings, T5-style relative bias, ALiBi, Rotary, and even removing PE) find large gaps on reasoning and math tasks, indicating that PE choice can either help or hinder extrapolation beyond the training lengths (Jelassi et al., 2023; Kazemnejad et al., 2023; Press et al., 2022; Su et al., 2023; Li et al., 2024; Golovneva et al., 2024). Randomized positional encodings address this by sampling from the distribution of longer positions during training, which improves accuracy across a broad variety of algorithmic tasks (Ruoss et al., 2023).

**Data formatting and supervision signals.** Formatting the data to better fit next-token prediction (NTP) is another effective lever. Reversing the output for addition (LSB-first) turns the problem into a local per-position mapping that uses only two digits and the carry, which improves both accuracy and sample efficiency, which has become a well know techniques (Lee et al., 2023). Chain-of-thought reveals intermediate digit sums and carries can further reduce the number of training examples needed, although the benefit depends on how the steps are designed (Nye et al., 2021; Hou et al., 2024). Training and inference with pause tokens (Goyal et al., 2024) aim to elicit implicit chain-of-thought without CoT annotations. Random Spacing can give short-range gains (e.g., 10→12 digits), but it also disrupts the stable right-aligned correspondence between matching digits and can hurt indexing, limiting robustness (Shen et al., 2023).

**Explicit position markers and digit alignment.** A complementary direction is to expose digit significance directly in the input. Index Hints add symbolic markers that align corresponding digits across operands, which reduces the need for the model to perform index arithmetic and yields large gains on addition and other tasks under the RASP-L lens (Zhou et al., 2023). Beyond hints, both Position Coupling and Abacus Embeddings can be seen as embedding-level position markers that enforce or learn a consistent alignment for digits of equal place value (Cho et al., 2024; McLeish et al., 2024). Predictive Position Coupling further extends the coupling ideas to handle input-dependent alignment needs (Golowich et al., 2025).

**Advanced Training and Architectural Strategies** Advanced training paradigms and architectural modifications also contribute to length generalization. These include Rule-Following Fine-Tuning (Hu et al., 2024), Attention Bias Calibration (Duan et al., 2024), and iterative self-improvement frameworks (Lee et al., 2025). Architecturally, Looped Transformers (Fan et al., 2025; Dehghani et al., 2019) improve generalization on iterative tasks by enabling adaptive processing steps.

**Theory and diagnostic frameworks.** RASP-L provides a predictive framework: if a task admits a short RASP-L program that works for all lengths, standard decoder-only Transformers trained to completion tend to learn a solution that length generalizes; if not, they usually fail (Zhou et al., 2023). This view helps explain why counting, sorting, unique-copy, and mode often extrapolate, while vanilla decimal addition and parity do not—unless the format or supervision makes the underlying program easier for a Transformer to represent (e.g., reverse output, index hints).

## B    FORMAL DEFINITION OF ALIGNED BLANKSPACE AUGMENTATION

### B.1    ABA ALGORITHM

The full ABA routine is in Alg. 1; its helper for inserting shared spaces is in Alg. 2. Helper functions are defined below:

**ParseEquation**$(E) \rightarrow (\mathcal{C}, \mathcal{O}ps)$. Given an equation string $E$ (digits, operators, and '='), return the ordered list of numerical components $\mathcal{C}$ (each as a string) and the operator/structure tokens $\mathcal{O}ps$ that record the type of operation.

**ZeropadLength**$(L_{\mathrm{op}}, \mathcal{O}ps) \rightarrow L$. Given the maximum operand length $L_{\mathrm{op}}$ and the operator pattern, return the base zero-padding length. Examples to set $L$ based on the operand is in Tab. 3

**LeftPad**$(s, L, \mathtt{'0'}) \rightarrow \tilde{s}$. Left-pad string $s$ with the character $\mathtt{'0'}$ until its length is $L$.

**RandomInteger**$(\mathrm{from} = a, \mathrm{to} = b) \rightarrow r$. Sample an integer $r$ uniformly from $\{a, a+1, \ldots, b\}$ (inclusive).

**Add_Synchronized_Spaces**$(\widetilde{\mathcal{C}}, p) \rightarrow \mathcal{C}'$ (Algorithm 2). Given zero-padded components $\widetilde{\mathcal{C}}$ of common length $L$ and a target length $p > L$, choose $L$ shared positions in $\{0, \ldots, p-1\}$ and place the $k$-th digit of every component at the same chosen index; all other positions are the blank token $\square$.

**Reconstruct**$(\mathcal{C}', \mathcal{O}ps) \rightarrow E'$. Merge the augmented numerals $\mathcal{C}'$ with the operator/structure template $\mathcal{O}ps$ to form the final equation string $E'$.

**CreateArray**$(\text{size} = p, \text{fill} = \square) \rightarrow \mathbf{z}$. Return a length-$p$ array initialized with the blank token $\square$.

**RandomSample**$(\mathbf{I}, \text{count} = L, \text{without\_replacement} = \texttt{true}) \rightarrow \mathbf{D}$. Uniformly sample $L$ distinct indices from index set $\mathbf{I}$.

**SortAscending**$(\mathbf{D})$. Sort $\mathbf{D}$ in increasing order (in-place).

**Join**$(\mathbf{z}) \rightarrow s$. Concatenate the entries of array $\mathbf{z}$ to a single string.

---

**Algorithm 1** Generate_ABA_Example

---

**Require:** Equation string $E$, maximum target length $p_{\max}$, boolean $is\_fixed\_variant$ ($True$ for ABA-fixed, $False$ for ABA-var).
**Ensure:** Augmented equation string $E'$.
1: $(\mathcal{C}, \mathcal{O}ps) \leftarrow \text{PARSEEQUATION}(E)$
2: $L_{\text{op}} \leftarrow \max\big(\text{len}(c) : c \in \mathcal{C}\big)$
3: $L \leftarrow \text{ZEROPADLENGTH}(L_{\text{op}}, \mathcal{O}ps)$
4: $\widetilde{\mathcal{C}} \leftarrow [\,]$
5: **for** each $c \in \mathcal{C}$ **do**
6:    $\widetilde{c} \leftarrow \text{LEFTPAD}(c, L, \texttt{'0'})$
7:    append $\widetilde{c}$ to $\widetilde{\mathcal{C}}$
8: **end for**
9: **if** $is\_fixed\_variant$ **then**
10:    $p \leftarrow p_{\max}$
11: **else**
12:    $p \leftarrow \text{RANDOMINTEGER}(\text{from} = L + 1, \text{to} = p_{\max})$
13: **end if**
14: $\mathcal{C}' \leftarrow \text{ADD\_SYNCHRONIZED\_SPACES}(\widetilde{\mathcal{C}}, p)$          (uses Alg. 2)
15: $E' \leftarrow \text{RECONSTRUCT}(\mathcal{C}', \mathcal{O}ps)$
16: **return** $E'$

---

**Algorithm 2** Add_Synchronized_Spaces (helper)

---

**Require:** A list $\mathcal{N}$ of zero-padded numerical strings (same length), target length $p$.
**Ensure:** A list $\mathcal{A}$ of ABA-augmented strings.
1: $L \leftarrow \text{len}(\mathcal{N}[0])$
2: $\mathbf{I} \leftarrow [0, 1, \ldots, p-1]$
3: $\mathbf{D} \leftarrow \text{RANDOMSAMPLE}(\mathbf{I}, \text{count} = L, \text{without\_replacement} = \texttt{true})$
4: $\text{SORTASCENDING}(\mathbf{D})$
5: $\mathcal{A} \leftarrow [\,]$
6: **for** each $s \in \mathcal{N}$ **do**
7:    $\mathbf{z} \leftarrow \text{CREATEARRAY}(\text{size} = p, \text{fill} = \square)$
8:    **for** $k \leftarrow 0$ **to** $L - 1$ **do**
9:       $d \leftarrow s[k]$
10:       $j \leftarrow \mathbf{D}[k]$
11:       $\mathbf{z}[j] \leftarrow d$
12:    **end for**
13:    append $\text{JOIN}(\mathbf{z})$ to $\mathcal{A}$
14: **end for**
15: **return** $\mathcal{A}$

---

Table 3: **Zero-padding length $L$ used before inserting aligned blanks.**

| Task | Operands | Definition of $L$ |
|---|---|---|
| Addition | Two operands $(o_1, o_2)$ | $L = \max\{|o_1|, |o_2|\} + 1$ |
| Multi-operand Addition | $M$ operands $(o_1, \ldots, o_M)$ | $L = \max_i |o_i| + 1$ |
| $N \times 2$ Multiplication | $N$-digit $\times$ 2-digit | $L = |o_1| + |o_2|$ |
| General $N \times M$ Multiplication | $N$-digit $\times$ $M$-digit | $L = |o_1| + |o_2|$ |
| Sorting | $K$ numbers | $L = \max_i |o_i|$ |
| Copy / Reverse | Single string $u$ | $L = |u|$ |
| Caesar Cipher/String Reverse | Single string $u$ | $L = |u|$ |

## B.2 TASK SPECIFIC RULE TO DETERMINE THE ZEROPAD LENGTH $L$

Tab. 3 summarize the Zeropad length before inserting aligned blanks in the tasks introduced in the main text and appendix: Addition and Multi-operand Addition (Sec. 4.3), N×M Multiplication with scratchpad (Sec. 5), Sorting (Sec. 4.3), and Copy/Reverse with repeated tokens (Sec. 4.3) and Caesar Cipher/String Reversal (Appx. D.3.3). We note that length $L$ is determined to be sufficiently long to accommodate the longest operand and the expected maximum length of the sum.

## B.3 ABA EXAMPLE

We provide illustrative examples of our ABA-fixed and ABA-var strategies below, showcasing typical instances from both their respective training and test sets. The dataset is from 1-10 addition, and $p_{max}$ is set to 21. Note that the operands and answers in the equation are reversed.

```
 ABA-fixed Examples

 Training Set:
 $    1    28    0    000 +    3    39    2    710 =    4    57    3    710 $
 $    8 45    97    55 70+   2 55    21    00 00=   0 01    29    55 70$
 $5    954 6 249    890    +1    355 5 555    300    =6    210 2 894    201    $
 $    8 7 8    1    17    0 +   7 0 0    0    00    0 =   5 8 8    1    17    0 $
 $    1 5    0    542 0+    1 2    6    000 0=    2 7    6    542 0$
 $4    47 2 86    13 0    +9    70 0 00    00 0    =3    28 2 86    13 0    $
 $ 7 7    9    6    0 + 0 3    0    0    0 = 7 0    0    7    0 $
 $    95    81 29    2 0 +   89    70 00    0 0 =   75    62 29    2 0 $
 $ 9 2 00    0 00    0 + 5 6 32    1 64    0 = 4 9 32    1 64    0 $
 $9 539    41 68    0 0 +7 374    65 48    1 0 =6 904    17 07    2 0 $

 Test Set:
 $4680                     +9060                     =3741                          $
 $09899160                 +62265940                 =61165111                      $
 $701331550                +712304570                =423635031                     $
 $43474145980              +60862791270              =04247837161                   $
 $9065020                  +9493140                  =8559160                       $
 $31914113810              +43988215040              =74803428850                   $
 $9515635670               +2675389540               =1290025221                    $
 $3123550                  +6614190                  =9737641                        $
 $978830                   +737310                   =616250                         $
 $073510490                +623044340                =696554731                      $
```

```
 ABA-var Examples

 Training Set:
 $    595462 4989 0+    135555 5530 0=    621028 9420 1$
 $8    781    1    7 0+7    000    0    0 0=5    881    1    7 0$
```

```
$ 1 505420   + 1 260000   = 2 765420   $
$ 4    472861 3  0 + 9    700000 0  0 = 3   282861 3  0 $
$7     7 96  0+0     3 00  0=7     0 07  0$
$4856147230+5903832800=9769979040$
$ 9 58 1292  0 + 8 97 0000  0 = 7 56 2292  0 $
$9 2000000  +5 6321640  =4 9321640  $
$95394168  0 0+73746548  1 0=69041707  2 0$
$  05 1    4  92  8 0+  04 8    5  62  0 0=  09 9   9  55  8 0$

Test Set:
$166340+358530=415970$
$75213540+57499840=23703490$
$480+720=111$
$48970+82760=21741$
$86557920+18908230=94565260$
$72078710790+39186974460=02255794161$
$882380680+933131360=726412941$
$05518137720+30264458440=35772685270$
$1069801970+6234202020=7293113990$
$70+20=90$
```

## C  TRAINING CONFIGURATION AND COMPUTATIONAL ANALYSIS

In this section, we first detail the training configurations, including model setups and datasets used across our experiments. We then provide a quantitative analysis of the computational cost of ABA, measuring its impact on training time, memory usage, and inference efficiency under different model sizes.

### C.1  MODEL CONFIGURATION

The model configurations and hyperparameters used for our experiments are detailed below. Except for experiments for Sec. 4.2 Other positional embedding, for models trained with our ABA strategy, the hyperparameters are provided in Tab. 4. For experiments involving modern relative positional embeddings, and to ensure comparability with related work(Zhou et al., 2024; Kazemnejad et al., 2023), we utilized a specific hyperparameter set outlined in Tab. 5. For the Abacus Embedding baseline, we adopted the hyperparameter settings from McLeish et al. (2024), which are summarized in Tab. 6. For the Position Coupling baseline, we followed the setup described in Cho et al. (2024), with its hyperparameters listed in Tab. 7. Specifically, for Position Coupling models, we used a 3-layer architecture for addition tasks (consistent with Figure 1 in their work, which achieved their best results for that task) and a 4-layer architecture for other algorithmic tasks (as per Section 6 in their work).

Regarding certain specialized positional encoding parameters: for the Offset Randomization Hyperparameter ($k$) in Abacus Embedding and the max-position parameter in Position Coupling, we set their values to be the maximum digit length of the intended test set plus two. This approach aims to enable robust generalization across the full range of tested sequence lengths. For example, if a model is intended to be evaluated on addition problems with operands up to 200 digits, this respective parameter ($k$ or max-position) would be set to 202. We utilized the original public code repository provided by Cho et al. (2024) for the Position Coupling experiments. For Abacus Embedding, we integrated the implementation made available by McLeish et al. (2024) into our own codebase. Although we have made every effort to faithfully replicate baseline models for Abacus Embedding, potential subtle differences in our implementation details might exist that could contribute to variations in performance. For the definitive results and methodology of Abacus Embedding, we encourage readers to consult the original publication (McLeish et al., 2024).

We also noticed some performance variability when replicating the Position Coupling method. Cho et al. (2024) describe two addition setups: one using a 3-layer model trained on 1-30 digit operands and another using a 1-layer model trained on 1-20 digit operands, both tested up to 200 digits (cf.

their Figures 1 and 3, respectively). We adopted the 3-layer model in our experiments. However, our training procedure differed in checkpointing frequency: we evaluated and saved checkpoints every 1000 iterations, unlike the 100-step evaluation frequency reported in the original work. This procedural difference might partly explain any observed performance discrepancies. Our replication results for Position Coupling are presented in Fig. 8.

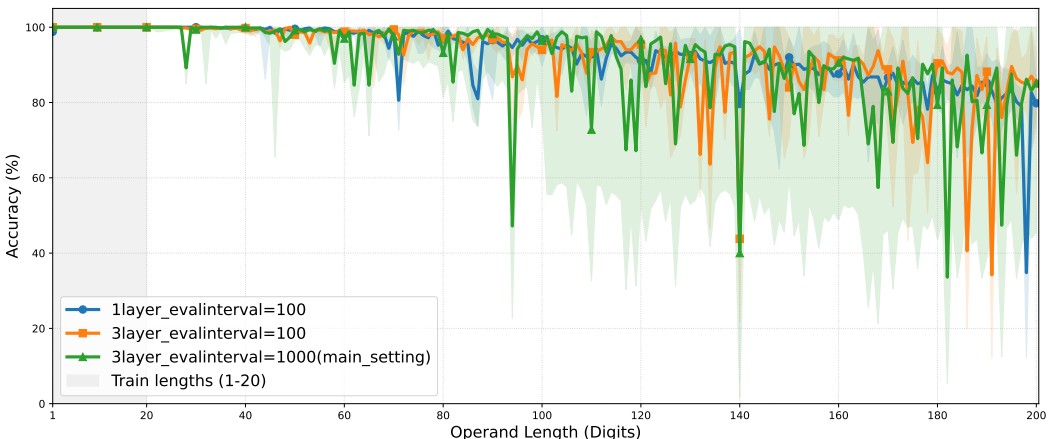

Figure 8: **Performance comparison of Position Coupling using different model size and different evaluation frequency.**

Table 4: **Hyperparameter for Model Using ABA**

| Hyperparameter | Default Value |
| --- | --- |
| Architecture | Decoder-only Transformer |
| Embedding Size | 384 |
| Number of Attention Heads | 6 |
| Number of Layers | 6 |
| Data Type | float16 |
| Weight Decay | 0.1 |
| Dropout | 0.2 |
| Optimizer | AdamW (Loshchilov & Hutter, 2019) |
| Global Batch Size | 512 |
| Learning Rate | 0.0005 |
| Learning Rate Scheduler | Cosine Decay (From LR to 0.1LR) |
| Feed Forward Layer | MLP (Vaswani et al., 2017) |
| Normalization Layer | LayerNorm (Ba et al., 2016) |
| Normalization Type | Pre |

## C.2 TRAINING SETUP

We adopted a standard causal language autoregressive training approach. When calculating the loss, we excluded the tokens corresponding to the input query (the problem statement), calculated only on the answer tokens and the EOS marker (denoted as $). All models were trained from scratch. Unless otherwise specified for a particular experiment, each model was trained for 50000 iterations. For models using Abacus embedding, due to its large model size and global batch size, we trained it for 20000 iterations. During training, checkpoints were evaluated every 1000 steps on a validation set to test their length generalization ability. The model checkpoint that achieved the highest exact-match accuracy on the validation set was selected for final evaluation. All reported test accuracies are the median over 5 independent runs with different random seeds. All experiments ran on a single NVIDIA H200 GPU.

Table 5: **Hyperparameter for Model Using ABA in Sec. 4.2 Other positional embedding**

| Hyperparameter | Default Value |
| --- | --- |
| Architecture | Decoder-only Transformer |
| Embedding Size | 512 |
| Number of Attention Heads | 8 |
| Number of Layers | 6 |
| Data Type | float16 |
| Optimizer | AdamW (Loshchilov & Hutter, 2019) |
| Global Batch Size | 512 |
| Weight Decay | 0.01 |
| Dropout | 0.0 |
| Learning Rate | 0.0001 |
| Learning Rate Scheduler | Cosine Decay (From LR to 0.1LR) |
| Feed Forward Layer | GatedGlu (Shazeer, 2020) |
| Normalization Layer | LayerNorm (Ba et al., 2016) |
| Normalization Type | Post |

Table 6: **Hyperparameter for Model Using Abacus**

| Hyperparameter | Default Value |
| --- | --- |
| Architecture | Decoder-only Transformer |
| Embedding Size | 1024 |
| Number of Attention Heads | 16 |
| Number of Layers | 16 |
| Data Type | float16 |
| Optimizer | AdamW (Loshchilov & Hutter, 2019) |
| Global Batch Size | 8192 |
| Weight Decay | 0.01 |
| Dropout | 0.0 |
| Learning Rate | 0.0001 |
| Learning Rate Scheduler | Cosine Decay (From LR to 0.1LR) |
| Feed Forward Layer | GatedGlu (Shazeer, 2020) |
| Normalization Layer | LayerNorm (Ba et al., 2016) |
| Normalization Type | Post |

Table 7: **Hyperparameters for Model Using Position Coupling**

| Hyperparameter | Default Value |
| --- | --- |
| Architecture | Decoder-only Transformer |
| Embedding Size | 1024 |
| Number of Attention Heads | 8 |
| Number of Layers | 3 or 4 |
| Data Type | float16 |
| Optimizer | AdamW (Loshchilov & Hutter, 2019) |
| Global Batch Size | 1000 |
| Weight Decay | 0.0 |
| Dropout | 0.0 |
| Learning Rate | 0.00003 |
| Learning Rate Scheduler | Cosine Decay (From LR to 0.1LR) |
| Feed Forward Layer | GatedGlu (Shazeer, 2020) |
| Normalization Layer | RMSNorm (Zhang & Sennrich, 2019) |
| Normalization Type | Pre and Post |

### C.3 DATASET

**Addition Task**    Our primary training dataset for addition comprises 1,000,000 examples. This dataset is intentionally imbalanced to give greater weight to longer operands within the specified training range (e.g., 1-10 digits for main experiments): the probability of sampling an $i$-digit problem (where both operands are of length $i$, or one is of length $i$ and the other is shorter) is proportional to $i$. For instance, in a 1-to-10 digit training set, 10-digit problems are sampled approximately 10 times more frequently than 1-digit problems. This sampling strategy helps mitigate an overabundance of shorter examples, which would disproportionately feature blank spaces under our ABA augmentation scheme. For baseline comparisons involving Abacus Embedding and Position Coupling, we used a uniform dataset distribution as specified in their respective original works, as we observed that our weighted sampling could negatively affect their performance.

For evaluating checkpoints during training (validation), we generated a validation set of 1,000 unique addition problems. In these problems, the operand length $N$ (for $N$-digit + $N$-digit additions) was sampled uniformly from the full range of lengths intended for out-of-distribution testing (e.g., 1 to 200 digits for experiments trained on up to 20-digit addition). Importantly, for both our ABA method and Position Coupling, equations are zero-padded to the maximum length of the two operands plus one. In contrast, Abacus Embedding does not employ zero-padding in our replication, consistent with its original paper's setup. For abacus and ABA, both operands and answers are reversed. For Position Coupling, only answers are reversed. For the accuracy vs. input length curve, we generated 500 unique problem instances for each specific operand length $N$ tested, where both operands have length $N$.

**Other Algorithmic Tasks**    For the other algorithmic tasks evaluated (Copy with Repeated Tokens, Reverse with Repeated Tokens, $N \times 2$ Multiplication, Sorting Task, and Multi-Operand Addition), training datasets comprised 100,000 examples per task. Operand digit lengths within these training sets were sampled using the weighted method described for the addition task, adapted to the specific training length range defined for each task (see main text, Sec. 4.3, for these ranges). For Multi-Operand Addition and the Sorting Task, the number of operands in the training set was sampled uniformly from 2 to 5. Validation sets for these tasks, used for evaluating checkpoints during training, consisted of 1000 samples each, with lengths and operand counts (where applicable) sampled uniformly from their respective test ranges. Again, for all tasks except Copy and Reverse, both operands and answers were zeropadded for our ABA method and for Position Coupling. ABA and abacus presented both operands and answers in reversed-digit order while Position Coupling only reversed the answers for these tasks, as consistent in their original setup.

### C.4 QUANTITATIVE ANALYSIS ON THE TRAINING AND MEMORY USAGE OF ABA

We experiment on 20-digit addition and test on 200-digit addition to measure the impact of ABA on training time and memory usage under varying model sizes and sequence lengths. We compare our ABA with $p_{\max}$ (which is the length of digits and blank spaces together) equal to 201 against a baseline without ABA augmentation across a model size of Small (6 layers, 384 embed) and Large (12 layers, 724 embed). Due to GPU memory constraints, we focused on this moderate $p_{\max}$ value.

In Tab. 8, the Training Iteration Time measures the total duration of a single training step for a batch of 500 samples on a single GPU. The Peak Training GPU Memory captures the maximum VRAM required for this process. The inference performance is assessed by the Average Latency, representing the end-to-end time needed to generate a complete response from a prompt. Peak Inference Memory indicates the maximum memory consumed during generation.

To provide fairness under a fixed training budget, we also measure how validation accuracy evolves as a function of training time. We train ABA-fixed+RPE with $p_{\max} = 201$ and Position Coupling on 20-digit addition and evaluate on 200-digit addition, using the same optimizer and batch size. Figure 9 shows that Position Coupling reaches high validation accuracy much earlier than ABA-fixed+RPE, while ABA-fixed+RPE only catches up after a much longer training time.

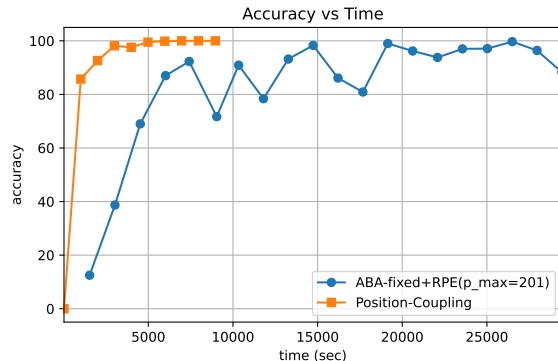

Figure 9: **Validation accuracy vs. training time** on 200-digit test for ABA-fixed+RPE ($p_{\max} = 201$) and Position Coupling.

Table 8: **Training and inference cost.**

| Configuration | Training Time (s/step) | Training Memory (GB) | Inference Latency (ms/step) | Inference Memory (GB) |
|---|---|---|---|---|
| Baseline (Small) | 93.887 | 1.289 | 1,698.08 | 0.758 |
| $p_{\max} = 201$ (Small) | 444.329 | 13.759 | 5,741.77 | 3.655 |
| Baseline (Large) | 197.82 | 5.413 | 3,114.83 | 1.529 |
| $p_{\max} = 201$ (Large) | 1,148.33 | 42.632 | 31,689.19 | 5.618 |

## D EXPERIMENT DETAILS

This section reports details of our experimental findings, evaluating how different methods generalize across multiple algorithmic tasks.

### D.1 COMPARISON WITH OTHER AUGMENTATION AND PADDING STRATEGIES IN ADDITION

This section presents additional results comparing our Aligned Blankspace Augmentation (ABA) method with the random space augmentation proposed by Shen et al. (2023) and the padding strategy introduced by Sabbaghi et al. (2024). We acknowledge that the original experimental setup of Shen et al. (2023) involved inserting random spaces within a scratchpad context, and Sabbaghi et al. (2024) utilized an encoder-decoder architecture; both differ from our decoder-only, non-scratchpad baseline configuration for these direct comparisons. Nevertheless, these methods offer valuable insights into data formatting for arithmetic reasoning.

To facilitate a focused comparison of the padding strategies themselves, all models were trained on 1-10 digit addition.

- When applying our **ABA** strategies (both ABA-fixed and ABA-var), the maximum component length $p_{\max}$ was set to 21.
- For the method by **Sabbaghi et al. (2024)**, each numerical component was padded with spaces to achieve a fixed total length of 21 tokens.
- For the **Random Spacing** method (Shen et al., 2023), spaces were inserted randomly into each numerical component such that its total length could vary but was capped at a maximum of 21 tokens.

Examples of the three methods are included in Appx. D.1. A comparative analysis of these approaches is presented in Fig. 10. Another interesting variant suggested by the reviewer during the rebuttal phase is to put blank spaces of random lengths on both the left and right sides of the operands. We include this format in Appx. D.1 and compare it against ABA in Fig. 10. Our findings indicate that both ABA-fixed and ABA-var generally surpass these alternative augmentation and padding methods in terms of length generalization performance on the addition task under these length-controlled conditions.

```
ABA:
$5  954 6 249   890   +1  355 5 555   300   =6  210 2 894   201   $

shen2023:
$   5 954 6 24 9 89 0+1355 5 555  300=62 10 2 894 201$

sabbaghi2024:
$59546249890          +13555555300          =62102894201          $

numbersInBetween:
$        59546249890     +     13555555300     =     62102894201     $
```

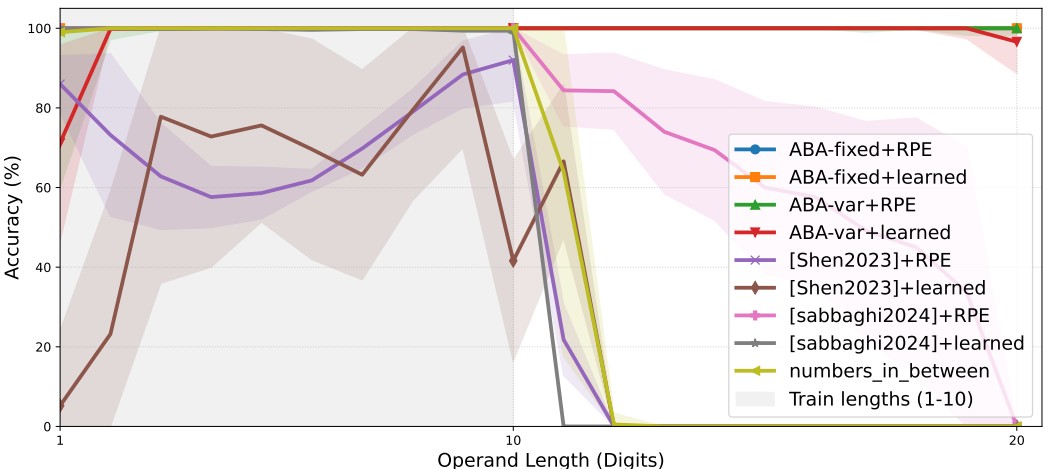

Figure 10: **Performance comparison of ABA (fixed and var strategies) against Random Spacing (Shen et al., 2023) and the padding strategy from Sabbaghi et al. (2024) on the addition task.** All models were trained on 1-10 digit addition and tested on longer operand lengths.

## D.2 SUPPLEMENTARY RESULTS FOR ADDITION UNDER DIVERSE CONDITIONS

This section provides supplementary figures and detailed data for the experiments on addition under diverse conditions, as presented in Sec. 4.2 of the main text.

### D.2.1 ADDITIONAL RESULTS ON 10+10 ADDITION

This subsection provides supplementary results for the Generalization from 10+10 experiment discussed in Sec. 4.2 of the main text.

**Ablations on Other Configuration.** We tested our four primary ABA configurations (ABA-fixed+RPE, ABA-fixed+learned, ABA-var+RPE, and ABA-var+learned) and Position Coupling and Abacus Embeddings. The heatmaps presented in Fig. 12 illustrate the exact-match accuracy across various test operand length combinations. For each cell in the heatmaps, performance was evaluated on 500 unique problem instances.

**Sequence-Matched Evaluation for Position Coupling.** ABA-fixed is evaluated in a setting where the total sequence length at test time matches the training length. Standard Position-Coupling does not have this property, which can confound comparisons. We therefore test whether Position-Coupling improves when its test-time sequence length is also matched to the training length. At test time we pad each operand on the left with zeros until its length is exactly 10 digits. This produces test inputs whose total sequence length matches the training configuration. We evaluate on operand lengths from 1 to 10 digits under this padded format. The result is shown in Fig. 11.

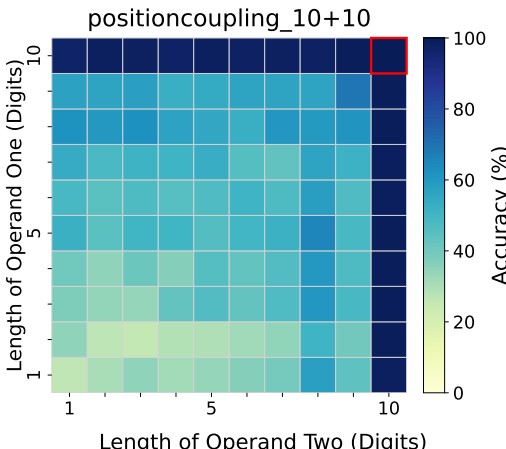

Figure 11: **Effect of sequence-length matching on Position Coupling. Position-Coupling is trained on 10-digit addition.** At test time, all operands are left-padded with zeros to 10 digits, so that the total sequence length matches the training length.

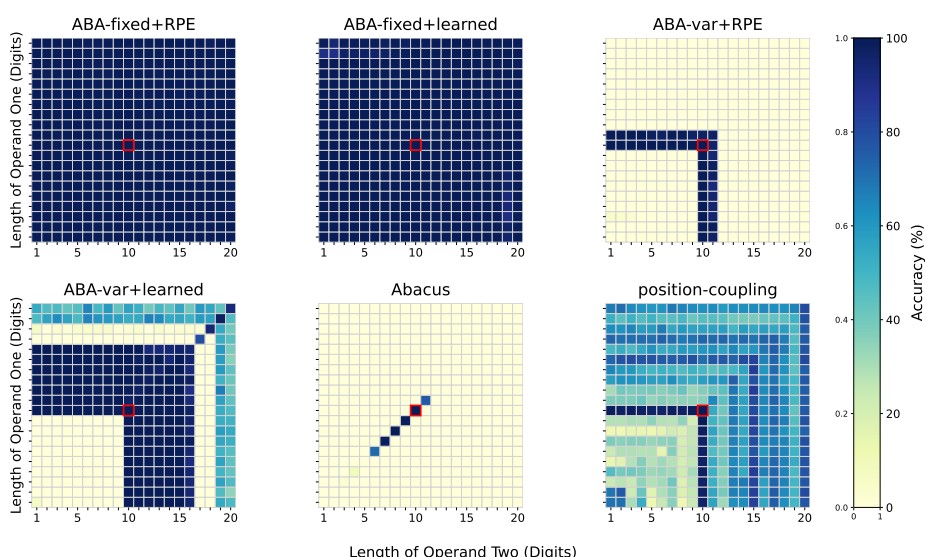

Figure 12: **Detailed performance heatmaps illustrating exact-match accuracy for models trained exclusively on 10+10 digit addition problems.** Each heatmap corresponds to a different approach: (a) ABA-fixed+RPE, (b) ABA-fixed+learned, (c) ABA-var+RPE, (d) ABA-var+learned, (e) Abacus Embedding, and (f) Position Coupling. Models were evaluated on operand pairs with lengths ranging from 1 to 20 digits (axes of heatmaps). Each cell reflects accuracy on 500 unique problem instances.

### D.2.2 ADDITIONAL RESULTS ON ABA-VAR FINETUNING

ABA-var uses the standard, space-free format at test time, which makes it natural to use as a finetuning scheme on top of a baseline model. We compare finetuning with ABA-var to training with ABA-var from scratch, in order to understand the tradeoff between compute cost and generalization. We consider addition with operand digits sampled uniformly from 1–10. In the *scratch* setting, we train a decoder-only Transformer directly on the ABA-var+RPE format with $p_{\max} \in \{31, 41, 51\}$, using the same architecture and optimizer as in Sec. 4.1. In the *finetune* setting, we first train a baseline model on the standard reversed zero-padded format, then reformat the same 1–10 digit dataset using

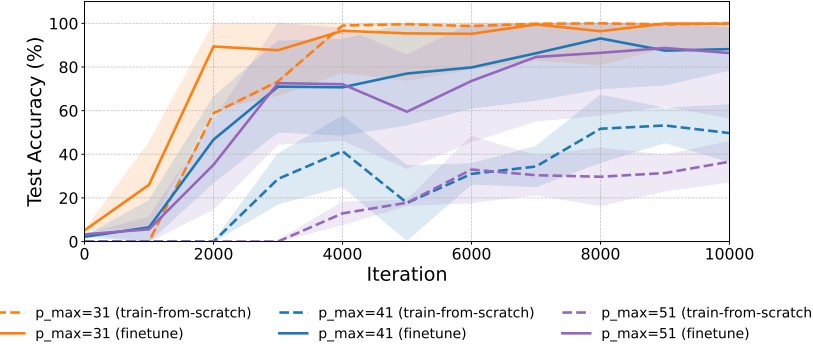

Figure 13: **ABA-var finetuning versus training from scratch.** Comparison of in-distribution (1–10 digits) and out-of-distribution (10–20 digits) accuracy for models trained with ABA-var+RPE either from scratch or by finetuning a baseline model. Finetuning converges faster for $p_{max} = 41$ and 51 while reaching similar final accuracy, showing that ABA-var finetuning is a compute-efficient way to transfer length generalization.

ABA-var and continue training from the baseline checkpoint with the same learning rate schedule. For both settings, we track and out-of-distribution accuracy (10–$p_{max} - 1$ digits) over training steps. The result shown in Fig. 13 demonstrates ABA-var finetuning as a compute-efficient alternative.

### D.2.3 ADDITIONAL RESULTS ON POSITIONAL EMBEDDINGS

This subsection presents detailed performance curves supplementing the other positional embedding experiments discussed in Sec. 4.2 in the main text, further illustrating how ABA-var and ABA-fixed interact with different positional encoding (PE) schemes. All models were trained on 1-10 digit addition and tested evaluated performance using 500 unique problem instances for each specific operand length $N$ (where both operands have length $N$). The results are depicted in Fig. 14.

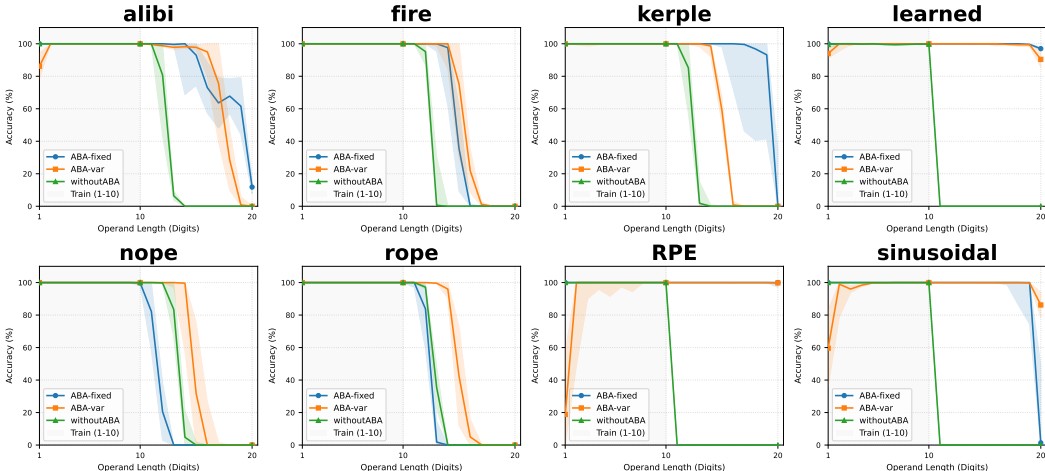

Figure 14: **Comparative length generalization performance for models using various positional embeddings (PEs) without ABA versus when combined with ABA-var and ABA-fixed.** The model are trained on the 1-10 digit addition task and tested up to 20 digits. Performance evaluated on 500 unique instances per test operand length.

### D.3 SETUPS AND SUPPLEMENTARY RESULTS ON ADDITIONAL ALGORITHMIC TASKS

This subsection provides detailed descriptions, experimental setups, and supplementary results for Sec. 4.3 in the main paper.

### D.3.1 TASK DESCRIPTIONS AND SETUPS

We evaluated ABA on several additional algorithmic tasks known for their length generalization challenges. The specific training and testing ranges for each are detailed below:

- **Copy with Repeated Tokens:** This task requires the model to exactly replicate a given binary input string. While copying unique tokens can allow for simpler generalization strategies, using repeated (binary) tokens presents a known challenge (Zhou et al., 2023; Huang et al., 2025). Models were trained on 1-10 digit binary strings and tested on lengths from 1 to 100.

- **Reverse with Repeated Tokens:** Similar to the Copy task, this task requires the model to take a binary input string and output its reversed sequence. Training was on 1-10 digit binary strings, with testing on lengths from 1 to 100.

- $N$**-digit** $\times$ **2-digit Multiplication** ($N \times 2$)**:** This task requires the model to multiply a variable-length $N$-digit number by a fixed 2-digit number, specifically testing generalization over the length $N$. Prior studies have indicated that Transformers often struggle with robust length generalization on such multiplication tasks (Duan et al., 2024; Jelassi et al., 2023). We trained on $N$ from 1-10 digits and tested on $N$ from 1 to 60. For applying Position Coupling and ABA to this task, the $N$-digit first operand and the resulting answer are each zero-padded to a base length of $N + 2$.

- **Sorting Task:** As introduced by McLeish et al. (2024), this task involves sorting numbers that are indirectly represented by alphabetical indices; the model outputs the sequence of indices corresponding to the numerically sorted original numbers. Their work highlighted challenges in generalization for this task, particularly concerning the number of operands. We trained with 2-5 operands, each with 1-5 digits, and tested on 2-10 operands, each with 1-10 digits. For Position Coupling and ABA, When applying Position Coupling and ABA to the Sorting Task, each number within a given problem instance is zero-padded to match the digit length of the longest number in that same instance.

- **Multi-Operand Addition:** Extending standard two-operand addition, this task requires computing the sum of a variable number of multi-digit integers. It was introduced by Cho et al. (2025), who showed that while methods like Position Coupling struggle with generalization on this task without a scratchpad, performance can be improved with one. For a fair comparison in our ABA evaluations, no scratchpads were used for any method on this task. We trained with 2-5 operands (1-5 digits each) and tested on 2-10 operands (1-10 digits each). For Position Coupling and ABA applications to Multi-Operand Addition, all input operands and the final sum are zero-padded to the maximum digit length among all input operands in a given instance plus one to accommodate potential carries in the sum.

The sampling of training and validation set is described in Appx. C.3. For the final test evaluation (results reported in Tab. 2 and other figures), test sets consisted of 10000 examples for each specific configuration tested. This means 500 samples per operand length for tasks like Copy, Reverse, and $N \times 2$-digit Multiplication, and 500 samples per combination of numbers of operands and operand length for the Sorting Task and Multi-Operand Addition. Digit lengths in these test sets were sampled uniformly from 1 up to the maximum tested digit length for that task. For Multi-Operand Addition and the Sorting Task, the number of test operands was sampled uniformly from 2 to 10.

### D.3.2 DETAILED RESULTS FOR OTHER ALGORITHMIC TASKS

This subsection presents the detailed length generalization performance for the tasks described above. For the Copy, Reverse, and $N \times 2$-digit Multiplication tasks, each tested data point (corresponding to a specific operand length $N$) consists of 500 unique examples. For the Sorting Task and Multi-Operand Addition, each tested data point in the heatmap (corresponding to a specific combination of digit length and number of operands also consists of 500 unique examples.

Note that for the test sets of Multi-Operand Addition and the Sorting Task, we adopted a specific sampling strategy to create challenging yet manageable instances: for a given target maximum digit length and a target number of operands, one operand was always set to this maximum digit length, while the lengths of the remaining operands were sampled uniformly from 1 up to this maximum

digit length. This approach was chosen to ensure test instances were not excessively difficult which might occur if all operands were always set to the maximum digit length.

The detailed results for each of these tasks are presented in Figures 15 to 19.

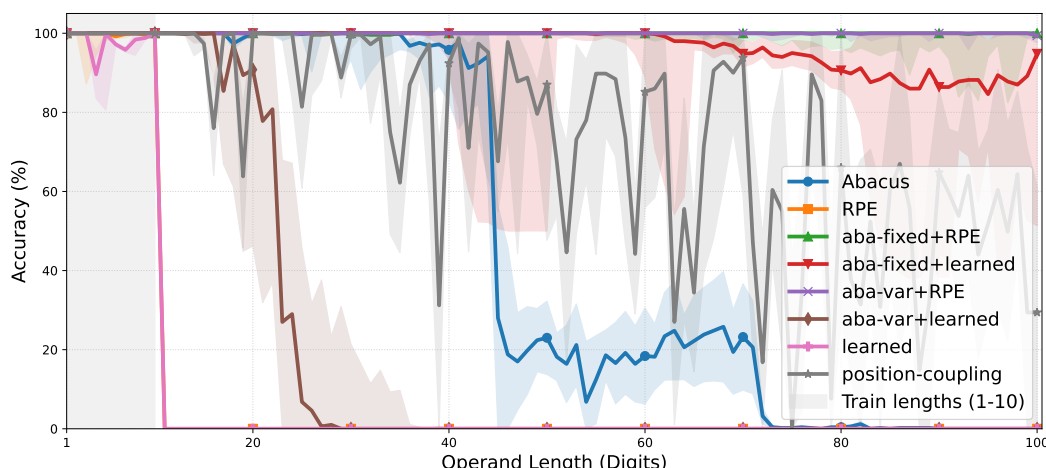

Figure 15: **Length generalization performance on the Copy with Repeated Tokens task.** Models were trained on 1-10 digit binary strings and tested on lengths from 1 to 100 digits.

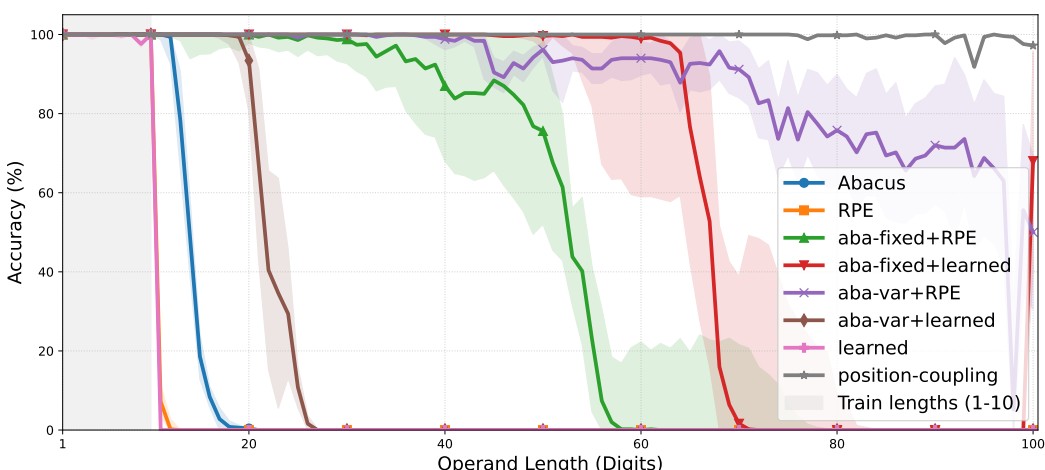

Figure 16: **Length generalization performance on the Reverse with Repeated Tokens task.** Models were trained on 1-10 digit binary strings and tested on lengths from 1 to 100 digits.

### D.3.3 ABA ON STRINGS

To check whether ABA can help on a symbolic transformation beyond arithmetic tasks with numbers, we involve two tasks below:

**Caesar Cipher**  This task checks ABA on a symbolic per-position mapping. Given a lowercase string $u$, the target $v$ applies a shift-by-one map modulo alphabet size (e.g. abc->bcd). Training uses 1–10 characters and testing 1–50 characters.

**String Reversal**  This task checks ABA when the mapping is global rather than local. Given $u$ of length $n$, the target is $v = \text{rev}(u)$ (e.g. abc->cba). Training uses 1–10 characters and testing 1–50 characters. ABA implementation is the same as the copy task described in Sec. 4.3

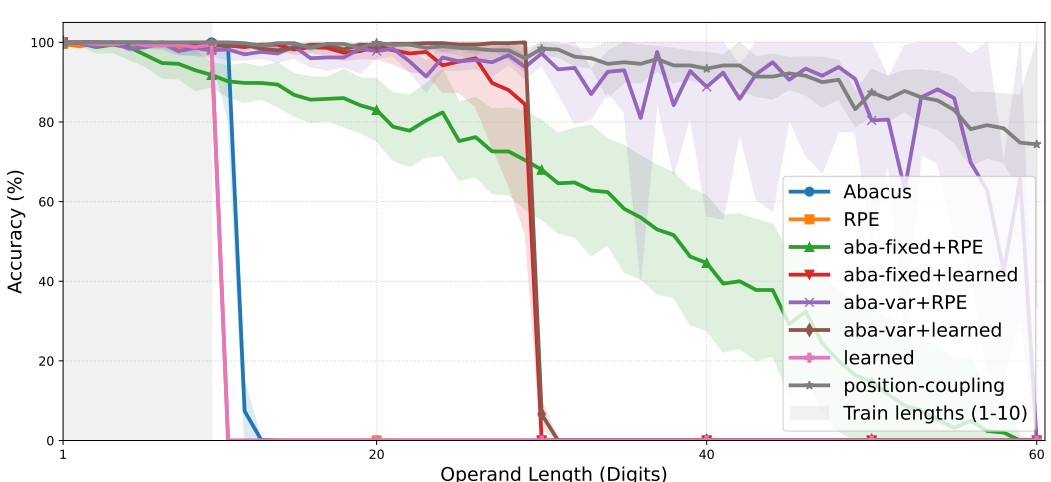

Figure 17: **Length generalization performance on the $N$-digit $\times$ 2-digit Multiplication task (generalizing over $N$).** Models were trained for $N$ from 1-10 digits and tested on $N$ from 1 to 60.

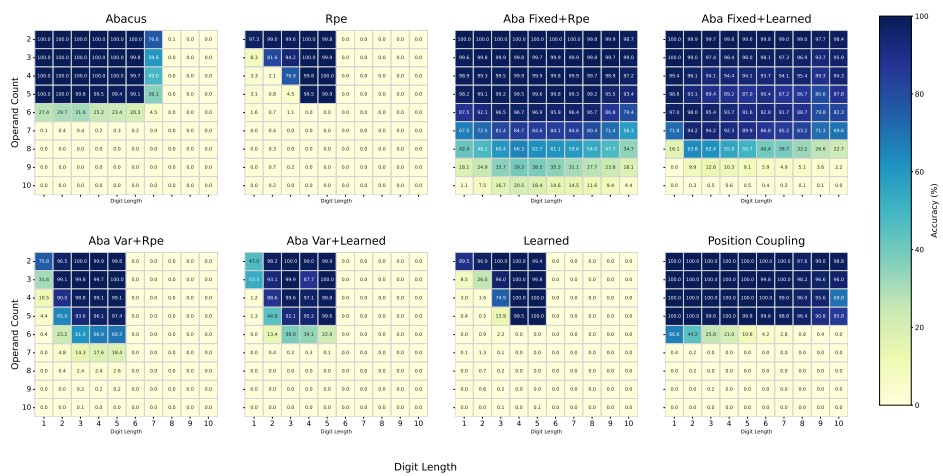

Figure 18: **Length generalization performance on the Multi-Operand Addition task.** Models were trained with 2-5 operands (each 1-5 digits long) and tested on combinations of 2-10 operands with digit lengths from 1 to 10.

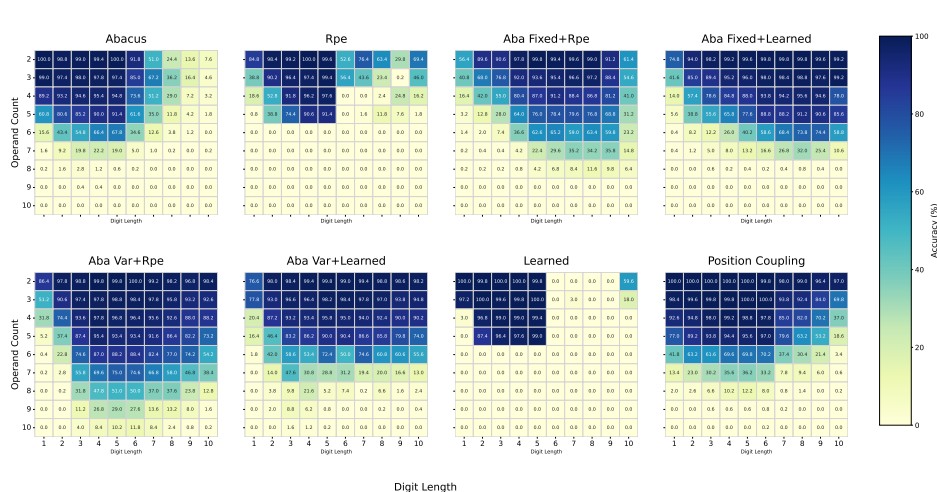

Figure 19: **Length generalization performance on the Sorting Task, where models sort numbers represented by alphabetical indices.** Models were trained with 2-5 operands (each 1-5 digits) and tested with 2-10 operands (each 1-10 digits)

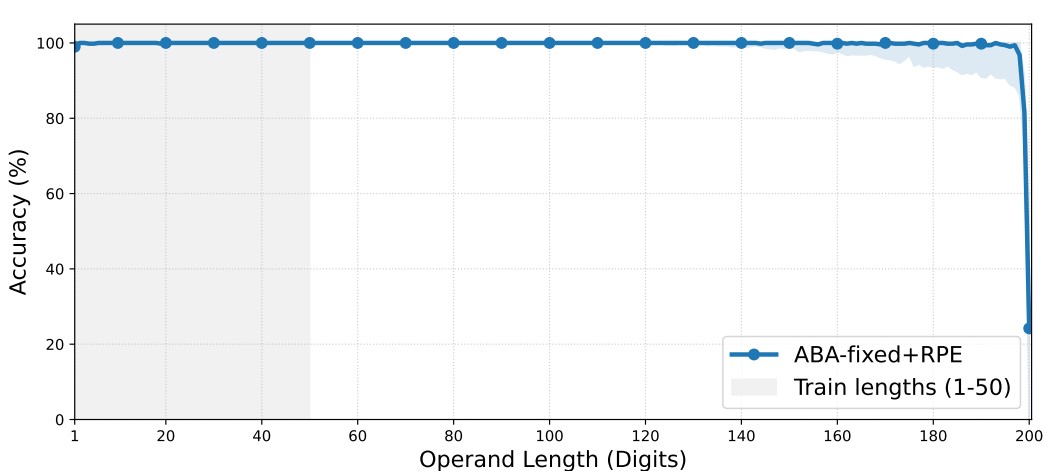

Figure 20: **Length generalization performance on the parity task.** Models were trained with each 1-50 digits and tested with each 1-200 digits.

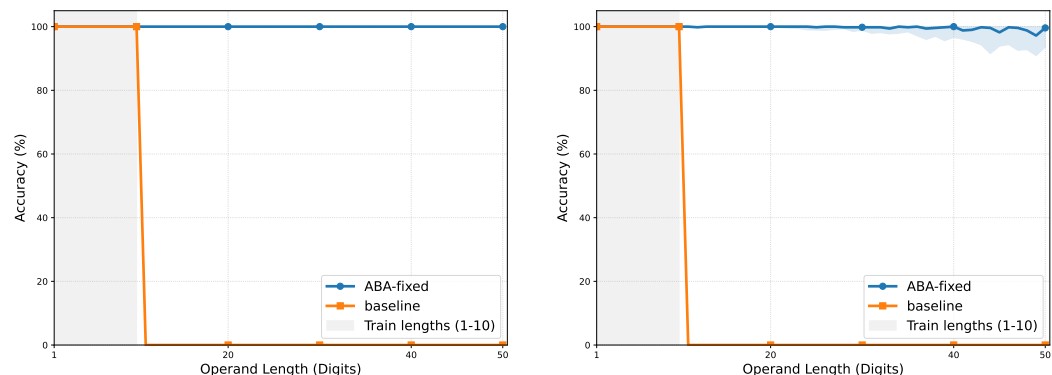

Figure 21: **Results on Caesar Cipher(left) and String Reversal(right).** Training uses 1–10 characters and testing 1–50 characters.

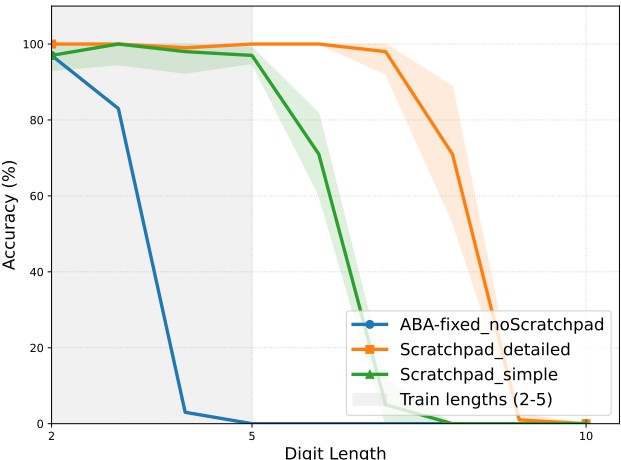

Figure 22: **Per-digit-Length test for multiplication.**

The result is shown in Appx. D.3.3. Across both Caesar and reversal, ABA consistently outperforms baselines, suggesting its potential in symbolic string tasks.

### D.3.4    PER-DIGIT-LENGTH TEST FOR MULTIPLICATION

To keep consistant with the other experiment, we include a per-digi-length test in addition to Fig. 7 for the baseline and the scratchpad method described in Sec. 5. For each operand length $l$, both the operands has the same length of $l$. The result is shown in Appx. D.3.4.

## E    ABLATION STUDY

This section presents ablation studies investigating the impact of model size (number of layers), training length (range of operand digits used for training), and the ABA hyperparameter. All ablations were conducted on the addition task.

### E.1    ABLATIONS ON MODEL SIZE

Fig. 23 compares the length generalization performance of models with varying numbers of layers (1, 2, 4, 6, and 8). For simplicity in this ablation, all models were configured with the ABA-fixed+RPE strategy, trained on 1-10 digit addition problems.

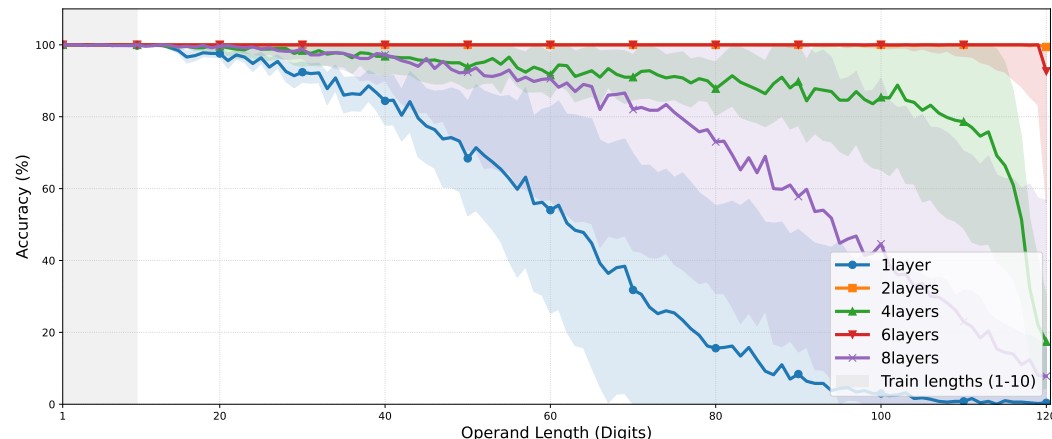

Figure 23: **Ablation study on model depth: Length generalization performance for models trained on 1-10 addition with varying numbers of layers (1, 2, 4, 6, and 8).** All models utilize the ABA-fixed strategy with RPE.

### E.2   ABLATIONS ON THE ABA PADDING-LENGTH HYPERPARAMETER $p_{\max}$

Fig. 24 illustrates how the length generalization performance of our four primary ABA configurations (ABA-fixed+RPE, ABA-fixed+Learned, ABA-var+RPE, and ABA-var+learned, all trained on 1-10 digit addition) varies with different settings of the ABA padding length parameter, $p_{\max}$. This study helps to understand the sensitivity of ABA to this crucial hyperparameter. We observe that overly large $p_{\max}$ can hurt length generalization and make training unstable. Our interpretation is that when $p_{\max}$ is set too high relative to the digit lengths in training, each sequence contains too few digits and a large fraction of blanks. This skews the token distribution toward spaces and weakens useful digit–digit interactions.

### E.3   ABLATIONS ON TRAIN LENGTH

Fig. 25 compares the impact of different training operand length ranges on length generalization. For each training length setting evaluated, the ABA parameter $p_{\max}$ was chosen heuristically based on insights gained from our 1-10 digit addition experiments to provide a reasonable configuration, with specific values indicated in the figure. While not exhaustively optimized for each training length, this approach allows us to observe the general trend.

To address the concern that ABA-fixed may be unscalable when the hyperparameter $p_{\max}$ is large, we search for a synergetic combination between training length and $p_{\max}$. We train ABA-fixed+RPE on addition with operand digit lengths up to 60, and $p_{\max}$ to 510. As before, ABA-fixed inserts aligned blank spaces so that the total length of each operand and the answer equals $p_{\max}$. After training, we evaluate the model on addition with operand lengths up to 500 digits and the result is shown in Fig. 26. Because the test lengths are very large, we sample evaluation points every 50 digits along the length axis. This experiment directly tests whether ABA-fixed can remain stable and generalize when both the training length and $p_{\max}$ are scaled up together. No

### E.4   ABLATIONS ON ALIGNMENTS

This task examines whether ABA requires strict structural alignment between digits. We introduce several modified training setups that disrupt digit-wise alignment to varying degrees. Models are trained on 1–10 digit addition and tested on 1–20 digit addition. $p_{\max}$ is set to 21 in this task.

The setups are as follows:

- **add_random_spaces:** Starting from ABA, random spaces are inserted into each operand. The number of inserted spaces is uniformly sampled from 0 to $\left\lfloor \frac{p_{\max}}{4} \right\rfloor$.

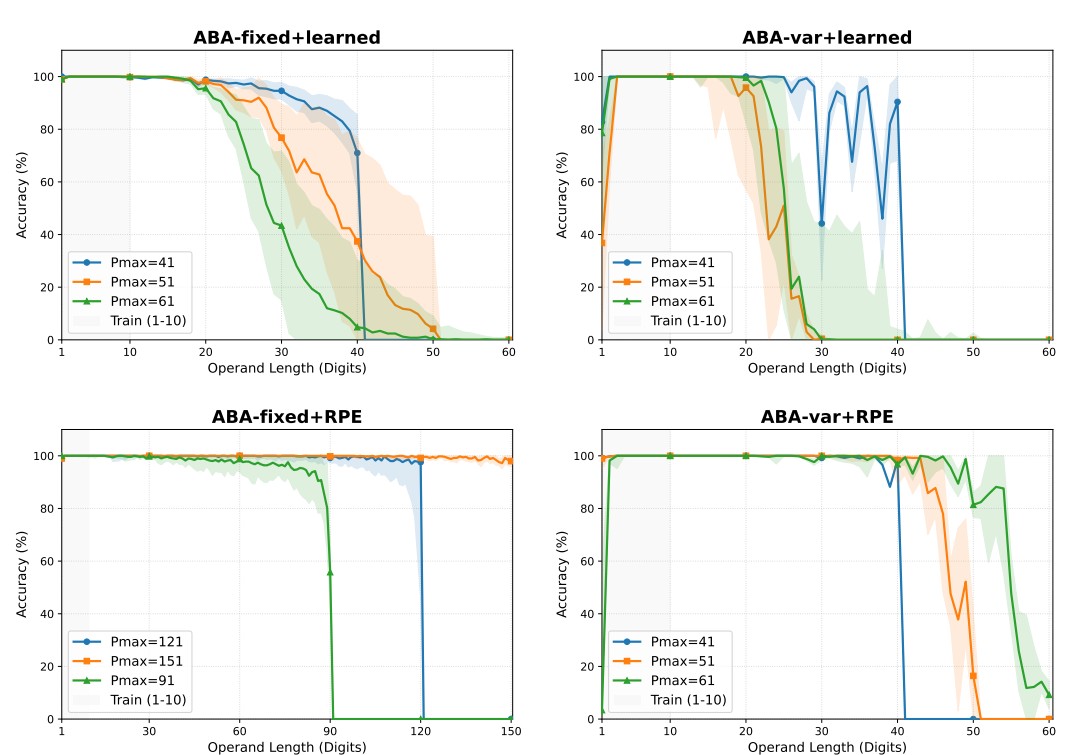

Figure 24: **Ablation study on the ABA padding length parameter** $P_{\max}$**: Performance of Models with ABA trained on 1-10 addition with different** $P_{\max}$ **values.**

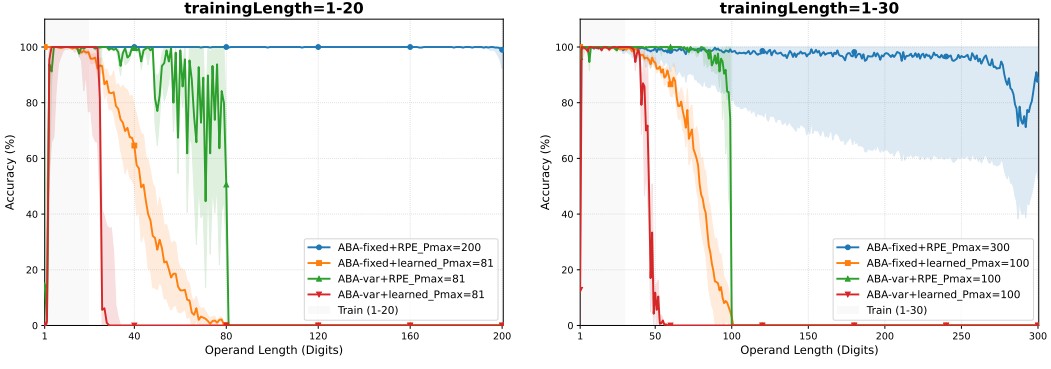

Figure 25: **Ablation study on training data length range: Performance of Models with ABA trained on 1-20 and 1-30 addition.**

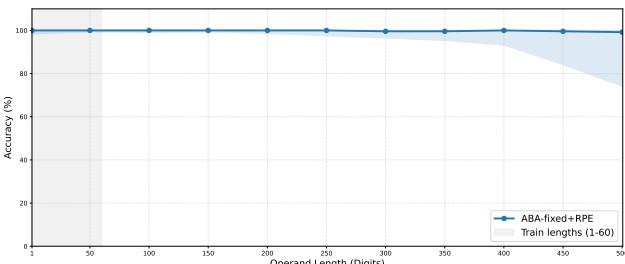

Figure 26: **Effect of training length and $p_{\max}$ on length generalization.** ABA-fixed+RPE is trained on 1–60 digit addition with $p_{\max} = 510$, then evaluated on up to 500-digit addition. The curve shows exact-match accuracy versus test digit length.

- **add_random_letter:** Starting from ABA, random letters are inserted into each operand. The number of inserted letters is uniformly sampled from 0 to to $\left\lfloor \frac{p_{\max}}{4} \right\rfloor$.

- **no_space_in_answer:** All spaces in the answer are removed, producing a continuous string of digits without column alignment.

- **reverse_b:** The second operand is written in reverse order while still occupying the same positions as in ABA. This preserves position alignment but reverses the digit order within the operand.

- **nozeropad:** Starting from the ABA format, we remove trailing blanks and zero padding on the right of each operand until the first non-zero digit; the same rule is applied to the test sets.

The result is shown in Fig. 27. The reverse_b setup achieves perfect 2× length generalization. The nozeropad condition with ABA-var also exhibits strong length generalization, and add_random_spaces combined with ABA-var retains some degree of generalization. These results indicate that ABA can tolerate a moderate amount of misalignment. We also observe that ABA-var generally outperforms ABA-fixed, suggesting that length variability helps the model withstand noise.

```
ABA:
$5 954 6 249 890 +1 355 5 555 300 =6 210 2 894 201 $
add_random_spaces:
$5 95 4 6 2 49 890 +1 3 55 5 5 55 30 0 =6 2 10 2 89 4 2 01 $
add_random_letter:
$5 95d4 6 24h9 890 +1 35g5 5r 555g 300 =6 2w10 2 8d94 2g01 $
no_space_in_answer:
$5 954 6 249 890 +1 355 5 555 300 =62102894201$
reverse_b:
$5 954 6 249 890 + 003 555 5 553 1=6 210 2 894 201$
nozeropad:
$5 954 6 249 89+1 355 5 555 3=6 210 2 894 201$
```

## F  SCRATCHPAD GENERATION DETAILS

This appendix details the generation process for the Aligned Blankspace Augmentation (ABA) integrated scratchpad used for the multiplication task, as discussed in Sec. 5. The primary goal is to decompose the multiplication into a sequence of simpler, aligned steps that mirror aspects of standard long multiplication, while consistently applying ABA-fixed to all numerical components to maintain structural integrity.

**General Formatting Principles for Scratchpad Numbers**    Let $op1_{orig}$ and $op2_{orig}$ be the two original integer operands for multiplication. Let $L_1$ and $L_2$ be their respective original digit lengths. The following principles guide the formatting of numbers within the scratchpad:

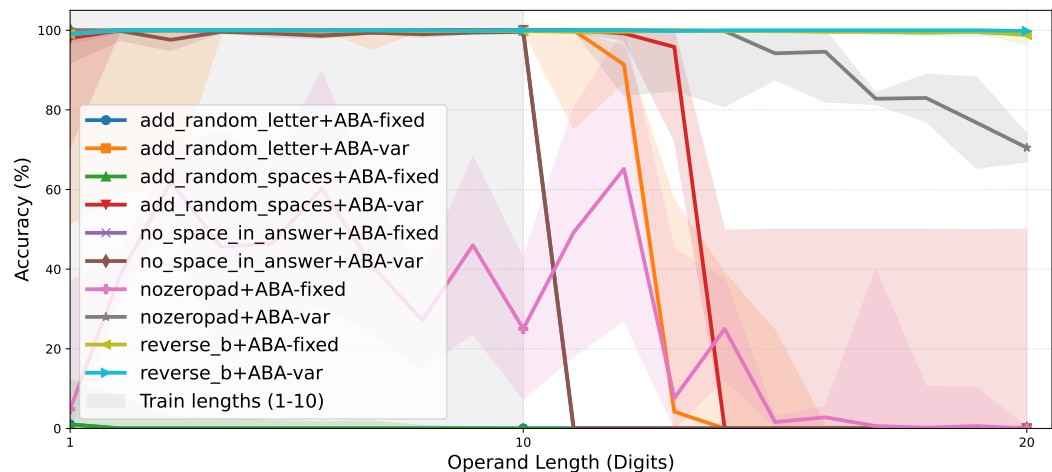

Figure 27: **Ablation study on alignment structure: Performance of Models with ABA trained on 1-10 addition.** All models in these experiments are with RPE.

1. **Formatting of Driving Operand (`spaced_op1`):** The first operand, $op1_{orig}$, is transformed to create `spaced_op1`, which dictates the scratchpad's iterative steps. The total number of tokens (digits plus spaces) in the resulting `spaced_op1`, is chosen within $L_1$ and $P'_{\max}$ (where $P'_{\max}$ is a predefined maximum target length for `spaced_op1`). The positions for these blank spaces within $op1_{orig}$ are also determined randomly.

2. **Zeropadded Length ($L_{num}$) for Other Numerals:** The second operand, $op2_{orig}$, and all intermediate numerical values generated in the scratchpad (such as partial products, running sums, and the final product) are first consistently zero-padded (typically left-padded) to a common base digit length, $L_{num}$, which is set to $L_1 + L_2$. This $L_{num}$ is chosen to accommodate the maximum possible digit length of the product of $op1_{orig}$ and $op2_{orig}$.

3. **ABA Formatting to Target Token Length ($P_{\max}$):** After being zero-padded to $L_{num}$ (as per point 2), $op2_{orig}$ (for its display in the header and use in calculations) and all key intermediate numerical values are then augmented with our ABA-fixed strategy formalized in Alg. 1 to target token length, $P_{\max}$.

4. **Number Reversal:** While illustrative examples in the main text present numbers in standard (non-reversed) order for simplicity, we reverse operands, intermediate values, and the final answer (digit-wise) within the scratchpad as we did in addition and other experiments.

For our experiments involving multiplication (e.g., training on operands up to 5 digits and testing up to 10 digits), we typically set $P_{\max} = 20$ to accommodate a maximum potential answer length of 20 digits and $P'_{\max} = 10$ for formatting $op1_{orig}$. We note that exploring larger $P_{\max}$ and $P'_{\max}$ values could potentially enable even stronger length generalization, though contingent on computational resources and training stability.

**Scratchpad Structure**   The scratchpad begins with a header row displaying the ABA-formatted multiplication problem (e.g., `$ spaced_op2 * spaced_op1 :`). The subsequent trace unfolds by iterating through the tokens (digits and spaces) of the ABA-formatted $op1_{orig}$ (i.e., `spaced_op1`), usually processed from its least significant token if reversal is active for training.

- For each **digit** in `spaced_op1`, a calculation step is generated. This involves computing a partial product, applying appropriate place-value shifting, and then presenting this (and potentially other intermediate values like a running sum) in ABA-formatted strings of length $P_{\max}$.

- For each **space** in `spaced_op1`, a corresponding placeholder row is generated, usually consisting of blank fields (also of length $P_{\max}$), to maintain the overall structural alignment dictated by `spaced_op1`.

The primary distinction between our two scratchpad variants lies in how these calculation steps are detailed:

DETAILED SCRATCHPAD (RUNNING SUM VARIANT): In this variant, each line corresponding to a digit from `spaced_op1` explicitly shows the current partial product, its place-value shifted version, the running sum from the previous step, and the new running sum, all ABA-formatted to length $P_{\max}$.

SIMPLIFIED SCRATCHPAD (ALIGNED PARTIAL PRODUCTS VARIANT): This variant omits the step-by-step accumulation of the running sum. For each digit from `spaced_op1`, it displays only the ABA-formatted, place-value shifted partial product. After processing all tokens from `spaced_op1`, a final line is constructed that explicitly presents the summation of all these stored and aligned shifted partial products, followed by the ABA-formatted final result of the original $N \times M$ multiplication. This effectively frames the core calculation as a multi-operand addition task.

Note that we use a reverse scratchpad in practice. The two operands and all intermediate results are reversed. We show an example of reverse scratchpad of 375*25321 below. We also show an prompt for testing (same for both detailed scratchpad and simplified scratchpad). Note operand1 is not padded, and blank spaces are inserted after operand 2 to reach $P_{\max} = 20$.

**Detailed Scratchpad:**

```
$5 73 *     12 3520 0   0  :
5*     12 3520 0   0  =     50 6621 0   0  >    50 6621 0   0  +    00 0000 0   0  =    50 6621 0   0  ,
 *     12 3520 0   0  =                     >                     +    50 6621 0   0  =    50 6621 0   0  ,
7*     12 3520 0   0  =     74 2771 0   0  >    07 4277 1   0  +    50 6621 0   0  =    57 0998 1   0  ,
3*     12 3520 0   0  =     36 9570 0   0  >    00 3695 7   0  +    57 0998 1   0  =    57 3594 9   0  ,
 *     12 3520 0   0  =                     >                     +    57 3594 9   0  =    57 3594 9   0 $
```

**Simplified Scratchpad:**

```
$57 3* 12 3    5 20   0  0:
5- 50 6    6 21   0  0
7- 07 4    2 77   1  0
 -
3- 00 3    6 95   7  0
> 50 6    6 21   0  0+ 07 4    2 77   1  0+                    + 00 3    6 95   7  0= 57 3    5 94   9  0$
```

**Test Prompt:**

```
$573*12352000            :
```

# G BACKGROUND ON COMMUNICATION COMPLEXITY AND LIMIT TRANSFORMERS

**Notation**: We present our notation for both Appx. G and Appx. H here. Let $\{0, 1\}^N$ be a bit string of length $N$. Similarly, let Let $\{0, 1, \_\}^N$ be a bit string of length $N$ which may include spaces which we denote by a character "$\_$". For a natural number $n \in \mathbb{N}$, we use the notation $[n] := \{1, ..., n\}$. Let $f : X \to Y$ represent a function with domain set $X$ and codomain set $Y$. Let $X \times Y$ represent the Cartesian product between $X$ and $Y$. For a vector $\boldsymbol{x} \in \mathbb{R}^d$, we denote by $x_{i,...,j}$ the entries from index $i$ to index $j$, $i \neq j$, $i, j = 1, ..., d$. We denote by $x_i$ the entry of $\boldsymbol{x}$ at index $i$. We denote by $D(f)$ the (worst-case, one-way) communication complexity of function $f$. For a function $f$, we denote $f \in \Omega(g)$ to represent that $\exists n \in \mathbb{N}, c \in \mathbb{R}^+$ s.t. for $n \geq N$, $f(n) \geq cg(n)$. Similarly, $f \in \mathcal{O}(g)$ means that $\exists n \in \mathbb{N}, c \in \mathbb{R}^+$ s.t. for $n \geq N$, $f(n) \leq cg(n)$. We denote $f \in \Theta(g)$ if $f \in \mathcal{O}(g)$ and $f \in \Omega(g)$.

**Communication Complexity**: First, we provide a brief background on communication complexity. Consider the function $f : \{0, 1\}^a \times \{0, 1\}^b \to \{0, 1\}$. We focus on "one-way communication complexity", where Alice communicates to Bob communicate one bit at a time using a particular *protocol*, with the goal of Bob successfully computing $f$; Bob does not send any bits to Alice. Communication complexity reasons about the number of bits required to be communicated by Alice and Bob to compute $f$ in the worst-case, over the best possible communication protocol. We denote this quantity by $D(f)$. Throughout Sec. 6, we refer to $D(f)$ as the communication complexity of a

problem. Note that Alice and Bob have unlimited "precomputation" and "postcomputation", i.e. only bits communicated from Alice to Bob, not Alice to Alice or Bob to Bob, count towards $D(f)$.

**Limit Transformers**: Limit Transformers Huang et al. (2025) are particular Transformers which provably length generalize over for a function or task $f$, constructed using an idealized inference process (differing from how Transformers are optimized in practice, like using SGD or ADAM). We denote a limit transformer as $T^*$. Huang et al. (2025) find that the existence of a limit transformer strongly corresponds to empirical length generalization.

**Communication Complexity and Limit Transformers**: In particular, Huang et al. (2025) prove that there exists a limit transformer $T^*$ that length generalizes over a function $f$ only if $D(f) \in \mathcal{O}(\log N)$, where $N$ is the input size to the problem. Thus, naive addition or naive copying (where tokens can be repeated), which have $\Omega(N)$ communication complexity, cannot be represented by a limit transformer.

## H PROOFS

### H.1 PROOF OF PROPOSITION 6.1

Please see Appx. G for notation. First, we state the following fact about addition with no augmentations:

**Proposition H.1.** *Suppose* $x \in \{0,1\}^{2K}$. *Suppose Alice has access to* $x_{1,\ldots,K}$ *and Bob has access to* $x_{K+1,\ldots,2K}$. *Then,* $f_{ADD} : \{0,1\}^K \times \{0,1\}^K \to \{0,1\}^{K+1}$ *which adds two $K$-length operands satisfies* $D(f_{ADD}) \in \Omega(K)$.

*Proof.* The proposition follows from a standard reduction from string equality, as discussed in Corollary 20 of Huang et al. (2025). □

In what follows, we restate Prop. 6.1 and provide a proof:

**Proposition H.2** (Formal Statement of Prop. 6.1). *Suppose* $x \in \{0,1\}^{2K}$. *Then, add spaces uniformly at random such that* $x' \in \{0,1,\_\}^{2p_{\max}}$, *where* $p_{\max} \geq K$. *Suppose Alice has access to* $x'_{1,\ldots,p_{\max}}$ *and Bob has access to* $x'_{p_{\max}+1,\ldots,2p_{\max}}$. *Then,* $f_{ADD\ SPACES\ UNALIGNED} : \{0,1,\_\}^{p_{\max}} \times \{0,1,\_\}^{p_{\max}} \to \{0,1,\_\}^{p_{\max}+1}$ *which adds two $p_{\max}$-length operands (which may have spaces inserted uniformly at random) satisfies* $D(f_{ADD\ SPACES\ UNALIGNED}) \in \Omega(p_{\max})$.

*Proof.* The case where $p_{\max} = K$ is trivial. Thus, consider the case that $p_{\max} > K$.

We first introduce the core concept of a *padding map*, which maps each index of the an input $y \in \{0,1,\_\}^{p_{\max}}$ to whether or not it is a space or not, i.e. $g : [p_{\max}] \to \{0,1\}^{p_{\max}}$, where 0 in the codomain of $g$ represents no space at that index and 1 in the codomain of $g$ represents a space at that index.

Next, note that in the setting of $f_{ADD\ SPACES\ UNALIGNED}$, Alice and Bob both can precompute a padding map which is *private* to Alice and Bob, i.e. $g_{Alice}$ and $g_{Bob}$. Such a padding map is private since Bob can only guess at Alice's padding map.

We now proceed to prove the proposition through a reduction from $f_{INDEX} : \{0,1\}^{p_{\max}} \times [p_{\max}] \to \{0,1\}$, where Alice has a string $x \in \{0,1\}^{p_{\max}}$ and Bob has an index $i \in [p_{\max}]$ and Bob must compute $x_i$. Note that $D(f_{INDEX}) \in \Omega(p_{\max})$ (Roughgarden, 2015). In particular, we demonstrate that any protocol for $f_{ADD\ SPACES\ UNALIGNED}$ can solve $f_{INDEX}$, and thus $D(f_{ADD\ SPACES\ UNALIGNED}) \in \Omega(p_{\max})$.

Let $\Pi$ be any protocol for $f_{ADD\ SPACES\ UNALIGNED}$. Suppose Alice has $x \in \{0,1\}^{p_{\max}}$. Alice precomputes strings $z^{(1)} := x_{1,\ldots,p_{\max}-1}$ and $z^{(2)} := x_{2,\ldots,p_{\max}}$. Alice then adds a single space uniformly at random to $z^{(1)}$ and $z^{(2)}$, yielding $z'^{(1)}, z'^{(2)} \in \{0,1,\_\}^{p_{\max}}$. Alice then sends Bob $z^{(1)}_1, z^{(2)}_2$, which can be done in $\Theta(p_{\max})$ bits. Bob now postcomputes a string of length $p_{\max} - 1$ of all 0s, i.e. $y := \underbrace{(0,\ldots,0)}_{p_{\max}-1 \text{ times}}$. Next, Bob further postcomputes by inserting a single space into $y$ uniformly at random, yielding $y$. Then, Bob further postcomputes by copying

$\boldsymbol{y}$ to yield $\boldsymbol{y}'^{(1)}, \boldsymbol{y}'^{(2)}$. Bob then runs $\Pi$ to yield $\boldsymbol{h}^{(1)} := f_{\text{ADD SPACES UNALIGNED}}(\boldsymbol{z}'^{(1)}, \boldsymbol{y}'^{(1)})$ and $\boldsymbol{h}^{(2)} := f_{\text{ADD SPACES UNALIGNED}}(\boldsymbol{z}'^{(2)}, \boldsymbol{y}'^{(2)})$. Bob finally postcomputes by concatenating $\boldsymbol{h}^{(1)}$ with $h^{(2)}_{2,\ldots,p_{\max}}$, recovering $\boldsymbol{x}$. Bob then takes his input index $i$ and return $x_i$, correctly computing $f_{\text{INDEX}}$. Thus, $D(f_{\text{ADD SPACES UNALIGNED}}) \geq D(f_{\text{INDEX}}) \in \Omega(p_{\max})$.

In particular, there cannot exist a limit transformer for the task $f_{\text{ADD SPACES UNALIGNED}}$ by Theorem 19 of Huang et al. (2025). □

**Discussion**: First, importantly, $f_{\text{ADD SPACES UNALIGNED}}$ considers adding two operands **of length $p_{\max}$, not of length $K$**. Furthermore, Alice's padding map is private and Bob's padding map is private, since spaces are added uniformly at random. Bob thus requires full knowledge of Alice's padding map, as he needs to know the positional information about all of Alice's bits as they align with his. Alice must thus send a full $p_{\max}$ length string. In particular, Alice and Bob cannot do precomputation to remove the spaces and then compute $f_{\text{ADD}}$, which would contradict the above proposition.

In addition, note that we consider $\boldsymbol{x} \in \{0,1\}^{2K}$ padded such that $\boldsymbol{x}' \in \{0,1,\_\}^{2p_{\max}}$ without loss of generality, as well as the codomain of $f_{\text{ADD SPACES UNALIGNED}}$ to include spaces without loss of generality; the above proof holds similarly in the case where each operand is only padded up to $p_{\max}$, rather than both padded to $p_{\max}$. To clarify, we assume that there are spaces in the output operand, and thus Alice cannot merely send the digits, but rather also needs to send their positions as well.

H.2    PROOF OF PROPOSITION 6.2

In what follows, we restate Prop. 6.2 and provide a proof:

**Proposition H.3.** *Suppose $\boldsymbol{x} \in \{0,1\}^{2K}$. Then, insert aligned spaces with ABA such that $\boldsymbol{x}' \in \{0,1,\_\}^{2p_{\max}}$ where $K \in \Theta(\log p_{\max})$. Suppose Alice has access to $\boldsymbol{x}'_{1,\ldots,p_{\max}}$ and Bob has access to $\boldsymbol{x}'_{p_{\max}+1,\ldots,2p_{\max}}$. Then, $f_{ABA} : \{0,1,\_\}^{p_{\max}} \times \{0,1,\_\}^{p_{\max}} \to \{0,1,\_\}^{p_{\max}+1}$ which adds two $p_{\max}$-length operands (which have spaces aligned between operands, inserted with ABA) satisfies $D(f_{ABA}) \in \mathcal{O}(\log p_{\max})$.*

*Proof.* In the case that spaces are added with ABA, rather than chosen uniformly at random, we design a simple $\mathcal{O}(K)$ communication protocol: since addition is being padded with ABA, rather than spaces being added uniformly at random, Alice and Bob share the same (public) padding map, i.e. $g_{\text{Alice}} = g_{\text{Bob}}$.[1] Thus, since Bob already knows Alice's padding map $g_{\text{Alice}}$, it suffices for Alice to send only the bits corresponding to the digits, where Alice precomputes where such bits are in her operand. Since Bob can precompute $g_{\text{Bob}} = g_{\text{Alice}}$, Bob can thus precompute where to put digits as they arrive from Alice. In particular, Alice sends $O(K)$ bits to Bob with any $O(K)$ protocol for $f_{\text{ADD}}$ e.g. bitwise.

Thus, there exists an $\mathcal{O}(K)$ protocol, $D(f_{\text{ABA}}) = D(f_{\text{ADD}}) \in \mathcal{O}(K) = \mathcal{O}(\log p_{\max})$. Therefore, in particular, there may exist a limit transformer for the task $f_{\text{ABA}}$ by Theorem 19 of Huang et al. (2025). □

**Discussion**: Note that Prop. H.2 holds for the more general problem of $p_{\max} \geq K$. However, $p_{\max}$ must be taken sufficiently large with respect to the original operand length size. This departs from our empirical results in Sec. 4, which demonstrate that operand lengths which scale linearly in the original input suffice. Still, this condition is required due to the limit Transformers framework, which states that having an $O(\log N)$ communication complexity is necessary for a limit transformer to exist; however, this is one particular transformer that length generalizes, obtained through an idealized inference procedure. Next, note that we do not necessarily require that $K \in \Theta(\log p_{\max})$ for this to hold; we only need that $p_{\max} \geq 2^K$. We however assume this for fair comparison with Prop. H.4 below.

Furthermore, we can conclude that Prop. H.2 and Prop. H.3 correspond exactly to explanation provided in the Sec. 1, which states that lacking alignment yields failure to length generalize.

Finally, note that we consider addition without loss of generality throughout both Prop. H.2 and Prop. H.3. For example, copying (without repeats) a $K$-length string also has a communication

---

[1] The concept of padding maps is discussed in Appx. H.1

complexity of $\Omega(K)$; we can show by the same methods as in the proofs of Prop. H.2 and Prop. H.3 that inserting unaligned spaces does not suffice to reduce communication complexity, while inserting aligned spaces does.

## H.3    STATEMENT AND PROOF OF PROPOSITION H.4

Notably, in Prop. H.3, we assume that $K \in \Theta(\log p_{\max})$. However, we assume in Prop. H.2 that $K \in \Omega(p_{\max})$. Still, for fair comparison, we provide a proposition below that characterizes the communication complexity for $K \in \Theta(\log p_{\max})$ as well. Note that communication complexity is reduced in this case; this is because $K \in \Theta(\log p_{\max})$ is a *special case* of the more general problem studied in Prop. H.2. This is similar to the fact that $n \times n$ matrix multiplication has a communication complexity of $\Omega(n^2)$, but $n \times n$ matrix multiplication *of diagonal matrices* has a communication complexity of $\Omega(n)$. Still, **our conclusions do not change, and random spaces does not admit a limit transformer, explaining why it fails to length generalize as well as ABA**.

**Proposition H.4.** *Suppose $\boldsymbol{x} \in \{0,1\}^{2K}$. Then, add spaces uniformly at random such that $\boldsymbol{x}' \in \{0,1,\_\}^{2p_{\max}}$, where $p_{\max} \geq 2^K$. Suppose Alice has access to $\boldsymbol{x}'_{1,...,p_{\max}}$ and Bob has access to $\boldsymbol{x}'_{p_{\max}+1,...,2p_{\max}}$. Then, $f_{ADD\ SPACES\ UNALIGNED} : \{0,1,\_\}^{p_{\max}} \times \{0,1,\_\}^{p_{\max}} \to \{0,1,\_\}^{p_{\max}+1}$ which adds two $p_{\max}$-length operands (which may have spaces inserted uniformly at random) satisfies $D(f_{ADD\ SPACES\ UNALIGNED}) \in \Omega(\log^2 p_{\max})$.*

*Proof.* We assume throughout that the reader is familiar with the padding maps introduced in the proof of Prop. H.2. Begin with an instance $\boldsymbol{x} = \underbrace{(0,...,0)}_{K \text{ times}}$ and augment it with an arbitrary padding map, i.e. we have $\hat{\boldsymbol{x}}$ with $K$ 0s and padding map $g$ arbitrary. Suppose this is Bob's operand, i.e. $\hat{\boldsymbol{x}}_B := \hat{\boldsymbol{x}}$. Without loss of generality, we consider the padding map has spaces after $K$ 0s, i.e. $\hat{\boldsymbol{x}}_B = (\underbrace{0,...,0,}_{K\text{times}} \underbrace{\_,...,\_}_{p_{\max}-K\text{times}} )$.

Next, $S$ be a set of possible operands for Alice, defined as:

$$S = \{\hat{\boldsymbol{x}}_A : \boldsymbol{x}_A \text{ is all 1s and } g_A \text{ maps the } K \text{ 1s to the first } 1, ..., p_{\max} - K \text{ positions}\}.$$

As such, $|S|$ is the number of ways to choose $K$ positions from $p_{\max} - K$ positions, i.e. $|S| = \binom{p_{\max}-K}{K}$.

Next, fix arbitrary $\hat{\boldsymbol{x}}_A^{(1)}, \hat{\boldsymbol{x}}_A^{(2)} \in S$, and note that $\boldsymbol{x}_A^{(1)} \neq \boldsymbol{x}_A^{(2)}$, where inequality is defined elementwise. Since, for this inequality to hold, positions of spaces must differ between operands, the positions of spaces in the outputs when passed through $f_{ADD\ SPACES\ UNALIGNED}$ with $\hat{\boldsymbol{x}}_B$ will be different. In particular, $f_{ADD\ SPACES\ UNALIGNED}(\hat{\boldsymbol{x}}_A^{(1)}, \hat{\boldsymbol{x}}_B) \neq f_{ADD\ SPACES\ UNALIGNED}(\hat{\boldsymbol{x}}_A^{(2)}, \hat{\boldsymbol{x}}_B)$. Since $\hat{\boldsymbol{x}}_A^{(1)}, \hat{\boldsymbol{x}}_A^{(2)}$ were arbitrary, this holds across all members of $S$.

Next, recall that a communication protocol between Alice and Bob must be able to generate $|S|$ different communication transcripts for $|S|$ different inputs for which the outputs differ. In particular:

$$D(f_{ADD\ SPACES\ UNALIGNED}) \geq \log |S| = \log \binom{p_{\max}-K}{K} = \log(\frac{p_{\max}-K}{K})^K).$$

since $\binom{n}{k} \geq (\frac{n}{k})^k$. By properties of log, this simplifies to

$$D(f_{ADD\ SPACES\ UNALIGNED}) \geq K(\log(p_{\max}-K) - \log K)).$$

For the first term, we have that:

$$\log(p_{\max}-K) = \log(p_{\max}(1 - \frac{K}{p_{\max}}) = \log p_{\max} - \log(1 - \frac{K}{p_{\max}}).$$

Since $K \in \Theta(\log p_{\max})$, $\frac{K}{p_{\max}} \to 0$ as $p_{\max} \to \infty$, yielding that $\frac{\log(1-\frac{K}{p_{\max}})}{\log p_{\max}} \to 0$ as $p_{\max} \to \infty$. As such, $\log(p_{\max}-K) \in \Theta(\log p_{\max})$, yielding:

$$D(f_{ADD\ SPACES\ UNALIGNED}) \in \Omega(K(\log p_{\max} - \log K))$$

.

Furthermore, since $K \in \Theta(\log p_{\max})$, $\frac{\log p_{\max} - \log K}{\log p_{\max}} \to 1$ as $p_{\max} \to \infty$. That is, $\log p_{\max} - \log K \in \Omega(\log p_{\max})$. Finally, since $K \in \Theta(\log p_{\max})$, we have that:

$$D(f_{\text{ADD SPACES UNALIGNED}}) \geq \Omega(K \log p_{\max}) \geq \Omega(\log^2 p_{\max}).$$

as desired.

Since $\log^2 p_{\max} \notin O(\log p_{\max})$, $f_{\text{ADD SPACES UNALIGNED}}$ still does not admit a limit transformer, thus explaining why it does not length generalize as demonstrated above. $\square$

**Discussion**: Importantly, the above proof relies on the fact that the first operand is padded. However, this is done without loss of generality; if only the second operand is padded, we switch the roles of Alice and Bob. Similarly to Prop. H.2 and Prop. H.3, addition is considered without loss of generality and all arguments follow similarly for tasks with linear worst-case communication complexity.

## I  LLM USAGE

We used a large language model (ChatGPT) as a writing assistant. Its role was limited to helping polish the language, improve clarity, and check grammar. All research ideas, experiments, analyses, and conclusions were conceived and executed entirely by the authors. The LLM did not contribute to research ideation, data analysis, or evaluation. The authors take full responsibility for the content of this paper.

