# OpenReview forum: "Data Augmentations for Arithmetic Length Generalization in Small Transformers"
_ICLR.cc/2026/Conference — Submitted to ICLR 2026_

### Official Review · Reviewer_2GoR · 2025-10-31

**Soundness:** 2
**Presentation:** 2
**Contribution:** 2
**Rating:** 4
**Confidence:** 3

**Summary:**

This paper presents a novel and compelling approach to the challenging problem of length generalization in Transformers for algorithmic tasks. The authors introduce ABA, a data augmentation technique that uses synchronized zero-padding and blank space insertion to enforce alignment between corresponding digits across operands. The central and powerful claim is that robust length generalization can be achieved without modifying the Transformer architecture. Also, the paper is supported by extensive experiments and a non-trivial theoretical analysis.

**Strengths:**

1. The proposed ABA method is remarkably simple yet effective, demonstrating that a purely data-centric intervention can outperform sophisticated architectural modifications like Position Coupling and Abacus Embeddings.
2. The experimental evaluation is thorough. The paper validates ABA's effectiveness not only on the canonical task of multi-digit addition, demonstrating generalization from 20 to 200 digits, but also on a diverse suite of algorithmic tasks including copying, reversing, sorting, and multi-operand addition.
3. The authors provide a principled explanation for why ABA works while random spacing does not. However, I have some questions about the proof.

**Weaknesses:**

See questions.

**Questions:**

1. ABA constructs training data by inserting spaces to extend the original numbers to a length of $p_{max}$. However, the paper mentions that the generalized length cannot exceed this constructed $p_{max}$. Since we're constructing data anyway, how would directly constructing data with longer operands perform? Would it be better than ABA? If so, could the authors explain the unique advantages of ABA? (Or are there other tasks where directly constructing longer data for training isn't feasible?)
2. In Table 2, ABA doesn't show a significant advantage compared to Position Coupling across the five tasks. Could this be related to the characteristics of the tasks, meaning some tasks are more suitable for enhancement using the ABA approach? Furthermore, the performance difference between the 'fixed' and 'var' variants is quite pronounced. What is the reason for this disparity? Is it due to inherent variability?
3. I didn't fully understand the application of ABA to the sorting task. Where should the newly added "operand placeholders" be placed in the answer? Also, in Figure 5, why does the raw data for the sorting task have three numbers (a, b, c) on the left side of the equals sign, but four on the right?
4. Regarding "Prop. 6.1", I believe the communication complexity should be at most $O(\log^2 p_{max})$. To communicate the padding map, one only needs to send the positions of K digits. Transmitting the position of a single digit requires $\log(p_{max})$ bits, so the total should be $O(K \log p_{max}) = O(\log^2 p_{max})$ bits, assuming $K = O(\log p_{max})$. The proof in the paper for "Proposition H.2" cites a conclusion from Roughgarden, 2015. That work doesn't have the $p_{max} >= 2^K$ restriction, so its worst-case scenario doesn't seem directly representative of the situation here.

---

> ### Author Response · Authors · 2025-11-22
> **Response to Reviewer 2GoR**
>
> We thank the reviewer for the careful reading of our paper and for the thoughtful questions, which helped us clarify both the practical scope and the theoretical aspects of ABA.
>
> ---
>
> >**Q1**:  Since we're constructing data anyway, how would directly constructing data with longer operands perform?
>
> **A1**: We believe this is an important question to clarify: Training directly on longer sequences does not solve the generalization issue. *Generalization **beyond** the training distribution is a fundamental question in machine learning*. Previous works [Lee et al.] have demonstrated that no matter how many digits you train the model in addition/multiplication, it fails to generalize to more digits. Many prior methods (e.g., those involving scratchpads or long padding) [Anil et al., Zhou et al., Shen et al.] also increase the sequence length and computational cost, yet their contribution lies in achieving better generalization, not efficiency.
>
>
> >**Q2**:  Some tasks are more suitable for enhancement using the position coupling approach?
>
> **A2**: Yes, we agree with the reviewer. This is a fundamental result in machine learning, which is a consequence of the “free lunch theorem”. Naturally, the inductive bias, i.e., the set of assumptions, play a critical role in the tasks that position coupling might emerge as beneficial. Few algorithmic tasks—particularly those with low positional sensitivity, such as copy or reverse—benefit less from ABA’s aligned structure. However, ABA remains valuable because it preserves the original model architecture, making it directly compatible with finetuning scenarios or settings where architectural modification (e.g., introducing custom position logic) is not feasible.
>
> >**Q3**: Elaborate the use of the placeholder and the extra ``d'' in the unaugmented data.
>
> **A3**:
> Thank you for pointing this out. The extra ``d'' that appeared in the original version of Figure 5 was a formatting error and is not part of actual data. We have corrected the figure in the revised manuscript so that the output sequence no longer contains this spurious symbol; the underlying sorting setup and results remain unchanged.
>
> The “operand placeholders” are appended to the answer sequence to represent potential future elements to be sorted. These placeholders enable the model to handle test cases with more items than seen during training
> .
> > **Q4**: In Proposition 6.1, the communication complexity of adding random spaces be at most $O(\log^2 p_{\max})$ if we assume that $p_{\max} \geq 2^K$?
>
> **A4**: We thank the reviewer for raising this important point. In our Proposition 6.1, we consider the more general case where $K \in O(p_{\max})$; in this case, the communication complexity is indeed at least $\Omega(p_{\max})$. However, like the reviewer mentions, if–as in our Prop 6.2 for ABA–we assume that $K \in \Theta(\log p_{\max})$, we obtain a communication complexity of at least $\Omega(\log^2 p_{\max})$ which matches the upper bound provided by the reviewer. This is **not a contradiction**, because the case where $p_{\max} \geq 2^K$ is a *restricted problem class* of the case where $p_{\max} \geq K$. To draw a parallel: when multiplying two $n \times n$ matrices, the communication complexity is at least $\Omega(n^2)$, but when multiplying two $n \times n$ *diagonal* matrice, the communication complexity is at least $\Omega(n)$.
>
> We have added this to our theory section in Sec. 6 to clarify this, as well as the proof of the lower bound when $K \in \Theta(\log p_{\max})$ as Prop. H.4 in Appx. H
>
> ---
>
> *Reference:*
>
> Anil, C., Wu, Y., Andreassen, A., Lewkowycz, A., Misra, V., Ramasesh, V., Slone, A., Gur-Ari, G., Dyer, E., & Neyshabur, B. (2022). Exploring length generalization in large language models. NeurIPS 2022.
>
> Sabbaghi, M., Pappas, G., Hassani, H., & Goel, S. (2024). Explicitly encoding structural symmetry is key to length generalization in arithmetic tasks. arXiv preprint, 2024.
>
> Shen, R., Bubeck, S., Eldan, R., Lee, Y. T., Li, Y., & Zhang, Y. (2023). Positional description matters for transformers arithmetic. arXiv preprint, 2023.
>
> Zhou, H., Bradley, A., Littwin, E., Razin, N., Saremi, O., Susskind, J., Bengio, S., & Nakkiran, P. (2024). What algorithms can transformers learn? A study in length generalization. ICLR 2024.
>
> **Final Note**:
> We are grateful for your detailed review and for the suggestions that helped improve this work. If you feel that our revisions and new experiments adequately address your concerns, we kindly ask you to consider raising your score and supporting the paper’s acceptance. Please do not hesitate to flag any remaining questions or doubts.

---

### Official Review · Reviewer_F1EX · 2025-11-02

**Soundness:** 3
**Presentation:** 4
**Contribution:** 2
**Rating:** 4
**Confidence:** 3

**Summary:**

The paper studies how to achieve length generalization in Transformers on arithmetic tasks.
While Transformers perform well on lengths seen during training, they often struggle on unseen lengths.
Prior work mainly improves length generalization through architectural changes or specialized positional encodings.
This paper proposes Aligned Blankspace Augmentation (ABA), a data augmentation strategy that inserts blank spaces at identical relative indices within operands and results.
With zero-padding and synchronized space insertion, the paper introduces two variants: ABA-fixed and ABA-var.
Using ABA-fixed with RPE, Transformers trained on addition with operand lengths up to 10 digits successfully generalize to up to 120 digits.
With ABA-var or with other positional encodings, the models still generalize, but less strongly overall.
The paper also applies ABA to other algorithmic tasks (Copying, Reversing, Multiplication, Multi-operand addition, Sorting).
With appropriate formatting (including placeholder operands), the trained models can generalize with respect to the number of operands.
Finally, the paper demonstrates length generalization on multiplication using a carefully designed scratchpad.

**Strengths:**

**S1.**
The paper is clearly written and provides thorough experimental details and comparisons to prior work.

**S2.**
The proposed method achieves strong length generalization across multiple tasks, especially in addition with ABA-fixed + RPE.

**Weaknesses:**

**W1.**
My major concern is that the proposed method requires exposing the model to long sequences during training.
Although the raw data are short, after augmentation the effective sequence length is set to a target $p$ that matches the test-time maximum via $p_{max}$.
This seems undesirable for a method intended to enhance length generalization.
If one has sufficient capacity to train on long sequences, why not simply train the model directly on longer sequences?


**W2.**
In Table 2, I was first surprised that ABA-fixed achieves length generalization for multi-operand addition task without a scratchpad, since the model has never added more than five single-digit numbers at once during training.
Even with placeholder operands, the model has not encountered sums with >5 single-digit addends during training, and thus I feel the result highly nontrivial.
For example, consider the problem 19+19+19+19+19+19+19=133.
To produce the correct result, the model should propagate a carry of 6 from the least-significant digit (9x7=63), whereas during training the largest carry from that column would be at most 4 (e.g., 9x5=45).
Therefore, I suspect that the proposed method may not faithfully length generalize for multi-operand addition task, especially where digits are concentrated in $\{8,9\}$ and carries become large.
Could the authors report results on such high-carry, high-operand-count problems (e.g., 6-10 operands with digits biased to {7,8,9})?
Testing longer per-operand digit lengths would be ideal, but for this stress test, keeping operand lengths within the training distribution is fine, as my concern here is whether the model can truly achieve generalization along the number of operands axis.

**Questions:**

**Q1.**
For the basic addition task, could the authors report accuracy specifically on long carry-chain cases (e.g., 999999999999999+1)?
This rare pattern can test whether the model truly learns carrying across the entire length.

**Q2.**
I don’t understand the sorting example in Figure 5.
Why does a $d$ appear in the output before augmentation (between Line 275 and 276)?

**Q3.**
In “Generalization from 10+10”, the paper claims ABA-fixed learns addition when trained only on 10-digit problems, whereas Position Coupling and Abacus fail.
In my personal view, the superior performance of ABA-fixed “primarily” stems from its methodological property that the training sequence length matches the test sequence length.
This is further reflected in ABA-var (Figure 9): the model generalizes to 16-digit inputs, but fails below 10 digits, since samples with $p < L (=10)$ never appear during training.
Therefore, I think this experiment does not strongly support the claimed superiority of the proposed method; the result appears highly trivial because the task is inherently advantageous for ABA-fixed.
I would like to hear the authors’ thoughts on this argument.

---

> ### Author Response · Authors · 2025-11-22
> **Response to Reviewer F1EX (Part I)**
>
> We thank the reviewer for the time and care devoted to reading our paper. The questions raised are thoughtful and highly valuable, and they helped us examine the limits and assumptions of our approach more carefully.
>
> >**Q1**: If one has sufficient capacity to train on long sequences, why not simply train the model directly on longer sequences?
>
> **A1**:  We believe this is an important question to clarify: Training directly on longer sequences does not solve the generalization issue. *Generalization **beyond** the training distribution is a fundamental question in machine learning*. Previous works, e.g., [Lee et al.],  have demonstrated that no matter how many digits you train the model in addition/multiplication, it fails to generalize to more digits. Many prior methods (e.g., those involving scratchpads or blankspace padding) [Anil et al., Zhou et al., Shen et al.] also increase the sequence length and computational cost, yet their contribution lies in achieving better generalization, not efficiency.
>
>
> >**Q2**: Could the authors report results on high-carry, high-operand-count problems?
>
> **A2**: We thank the reviewer for this insightful suggestion. This is indeed a valid concern, and we did not consider this possibility carefully in the first version.
>
> Following the reviewer’s suggestion, we conducted an additional evaluation on cases with both large carries and a larger number of operands. In this stress test, we generated sums with 1–10 operands, where each operand has 1–5 digits and every digit is sampled from {7,8,9}. This construction induces very large carries in every column.
>
> The result is in 4.4 (page8, figure6 middle). Although accuracy decreases as the number of operands increases, the model retains non-trivial accuracy across all operand counts across the entire range, and still reaches around 60% exact accuracy at 10 operands. This indicates that ABA-fixed can generalize across the number of operands, even when all columns induce carries that are substantially larger than anything seen during training.
>
> >**Q3**: Could the authors report accuracy specifically on long carry-chain cases?
>
> **A3**: Following the reviewer’s suggestion, we tested ABA-fixed on up to 10-digit addition with a full-carry test set, where the carry chain length equals the longer digit length (e.g., 9999999999 + 1) The result is in 4.4 (page8, figure6 left), the model achieves perfect accuracy on the full-carry test set.
>
> This supports our hypothesis that the model learns a digit-wise update rule for addition (and probably parity-scratchpad) that can be applied at any position seen in training, so performance does not degrade even when the carry chain spans the entire sequence. Together, these results suggest that the success of ABA is not limited to cases with short carry chains.
>
> >**Q4**: The extra ``d'' in Figure 5.
>
> **A4**:
> Thank you for pointing this out. The extra ``d'' that appeared in the original version of Figure 5 was a formatting error and is not part of actual data. We have corrected the figure in the revised manuscript so that the output sequence no longer contains this spurious symbol; the underlying sorting setup and results remain unchanged.
>
> >**Q5**: “Generalization from 10+10” is favorable to ABA-fixed.
>
> **A5**:  We agree with the reviewer that ABA-fixed benefits from the fact that its training and test sequences share the same overall length. To examine this point more directly, we performed an additional controlled experiment  where we modified the test set so that its test-time input length also matches its training length of 10 digits. Concretely, for any operand shorter than 10 digits, we pad it with leading zeros until the total sequence length is 10, making the evaluation setting symmetric to ABA-fixed.
>
> The result is in Appendix D.2.1. (page23, figure10). This adjustment indeed improves Position Coupling’s performance, especially for operand lengths 5–9, showing that part of its failure in the original setting comes from the mismatch between its training and test sequence lengths. However, even under this more favorable condition, Position Coupling does not reach perfect accuracy.
>
> These results support the reviewer’s point that matching sequence lengths helps Position Coupling, but they also suggest that sequence-length matching alone does not explain ABA-fixed’s superiority in this task.

---

> > ### Author Response · Authors · 2025-11-22
> > **Response to Reviewer F1EX (Part II)**
> >
> > *Reference:*
> >
> > Anil, C., Wu, Y., Andreassen, A., Lewkowycz, A., Misra, V., Ramasesh, V., Slone, A., Gur-Ari, G., Dyer, E., & Neyshabur, B. (2022). Exploring length generalization in large language models. NeurIPS 2022.
> >
> > Lee, Nayoung; Sreenivasan, Kartik; Lee, Jason D.; Lee, Kangwook; Papailiopoulos, Dimitris (2024). Teaching Arithmetic to Small Transformers. ICLR 2024.
> >
> > Shen, R., Bubeck, S., Eldan, R., Lee, Y. T., Li, Y., & Zhang, Y. (2023). Positional description matters for transformers arithmetic. arXiv preprint, 2023.
> >
> > Zhou, H., Bradley, A., Littwin, E., Razin, N., Saremi, O., Susskind, J., Bengio, S., & Nakkiran, P. (2024). What algorithms can transformers learn? A study in length generalization. ICLR 2024.
> >
> > **Final Note**: Thank you for your constructive feedback and careful assessment. We believe that the new analyses and edits substantially strengthen the paper, and we would greatly appreciate your consideration in revising your score and supporting acceptance. Should any part of our response be unclear, we would be glad to elaborate.

---

### Official Review · Reviewer_9M9W · 2025-11-03

**Soundness:** 1
**Presentation:** 3
**Contribution:** 2
**Rating:** 4
**Confidence:** 4

**Summary:**

This paper argues that a data augmentation technique called ‘Aligned Blankspace Augmentation (ABA)’ can improve the length generalization capability of Transformers in solving an arithmetic task. The ABA technique is summarized as a method of inserting ’blank spaces (a single non-digit non-operator token)’ between some digits of the operands and the answer, matching their relative positions. The blank space is used to match the trained sequence length to be (up to) the test length. For the integer addition task, the method is applied alongside a zero-padded, reversed format. The main method is compared with other state-of-the-art baselines, such as Position Coupling and Abacus Embedding, for length generalization in arithmetic tasks. As a result, the paper reports ‘ABA-fixed+RPE’ as their best method. The ABA strategy is also evaluated on other tasks such as multiplication (with/without scratchpads), copy/reverse, and more. Lastly, the paper also provides a communication-complexity-theory-based result explaining why the aligned blank spaces are beneficial in solving K-length addition with a limit transformer.

**Strengths:**

1. The paper is well-written. It clearly describes its main methodology, with great detail about the training and evaluation methods.
2. The paper provides abundant and quality experimental results to support their main claim, comparing with state-of-the-art training methods and data formats in the literature of transformer’s length generalization for arithmetic tasks.
3. The paper also provides a theoretical result to supplement their argument about the benefits of aligned blank spaces. (Honestly, I didn’t fully validate the proof yet. However, the statements are intuitively correct, at least in a high-level sense. Also, they provide meaningful insights on why the aligned blank spaces possibly enhance the trainability of a model towards a length-generalizable one. Hence, I put this point as a strength. I will investigate the proof after submitting this review.)

**Weaknesses:**

1. Speaking of ABA-fixed+RPE, which is the only baseline that beats all the existing methods (including Position Coupling and Abacus), I am afraid I cannot agree that this method is achieving the **length generalization**. Of course, I agree that the proposed training method is achieving a form of out-of-distribution generalization because the test data distribution is very different from the train data distribution (i.e., all the test instances lie outside of the support of the trained distribution). Nevertheless, these two distributions are identical in sequence length. In fact, I expect that ABA-fixed is not a length-generalizable method at all. Thus, I can only agree that ABA-var is achieving some form of length generalization, although it does not show a strictly better performance compared to the other baselines. Nonetheless, I think the authors can motivate their idea by taking ABA-fixed as an ‘oracle’ method (ideal-but-nonsense), instead of arguing it as their main methodology for length generalization.
2. The methods are not compared properly in terms of training cost. First of all, as admitted by the authors, the ABA strategy takes a much larger training cost than the other baselines because it requires training the model on (up to) target sequence length $p_{\rm max}$. Since the target length is often taken as ~10x larger than the training length (e.g., train up to 20, test up to 200), the computation cost for attention calculation must be up to ~100x larger in theory. Despite such a training inefficiency, a fair comparison between methodologies can be done if the training cost is matched strictly and manually (e.g., using a different number of steps or different mini-batch sizes to match the number of tokens seen during training or the training time). However, training of every baseline is run for 50K iterations (except for Abacus, which is trained for 20K iterations). This clearly indicates that the comparison in the paper violates the fairness in terms of training cost. Indeed, Table 8 showcases that ABA takes ~4.7—5.8x longer training time and ~7.9—10.7x larger training memory than the other baseline methods. With access to this large training budget, I believe that direct training on the target sequence lengths is possible and will achieve a much better performance.
3. The proposed method seems unscalable. This is because, as the authors discovered, an excessively large hyperparameter $p_{\max}$ hurts the training stability and hence causes failure in length generalization. Nevertheless, the paper provides a justifiable reason for this phenomenon, and I believe a similar issue will happen in other baselines. I would say it is expected that a stable training with a larger $p_{\max}$ requires longer training lengths. Hence, I would rather suggest finding a synergetic combination between the training sequence length and the hyperparameter $p_{\max}$ that can achieve a much larger generalizable length. As an example, Cho et al. (2024, Appendix B.1) have achieved a significant length generalization up to 500-digit additions by training a (small) model up to 160-digit additions.
4. Lastly, I am afraid I cannot agree with the authors’ argument that all the existing techniques require architectural modifications. In particular, the works by McLeish et al. (2024) and Cho et al. (2024) do not necessarily require a modification in the model code, given that the implementation already supports an input parameter for custom position IDs: see the explanation in Appendix C of Cho et al. (2024). The claim is partially true, for example, when we require multiple levels of position IDs (Cho et al., 2025). However, since the main subject of comparison is a simple integer addition, I do not think that ABA is a better methodology in terms of architectural modification: all three methods, Abacus, Position Coupling, and ABA, require only a modification in dataset construction.

**Questions:**

1. One thing that I was quite surprised about the proposed method is that it allows inserting blank space in the middle of the operands and the answer. One problem that I expected was that it might hinder computing and passing the carries. I am a bit suspicious that the success of ABA is because the carry chain is not very long in most of the examples generated uniformly randomly. In other words, I am curious whether the method is also successful for the examples with long carry chains or not.
    1. I would like to suggest a synthetic task that simulates the necessity of an extremely long carry chain: a parity task with a scratchpad. Given a binary sequence $(x_1, …, x_L) \in \Sigma^L$ as a prompt ($x_i \in \Sigma:=\\{0,1\\}$ for each $i$), the answer is a sequence $(y_1, …, y_L) \in \Sigma^L$ of the same length, defined as $y_i = \sum_{j=1}^{i} x_i \bmod 2 = x_i +y_{i-1} \bmod 2$ (letting $y_0 = 0$). Will the ABA method be successful in achieving length generalization for this task? How about a general $m$-parity task (letting $\Sigma = \\{0, 1, …, m-1\\}$ and taking $\bmod m$ instead of $\bmod 2$)?
    2. Believe it or not, exactly because of this reason, I have *personally* tried before an idea of putting the same number of zeros on the left/right of the operands and the answer to match the target length, although I am currently not planning to publish this idea as a conference paper. It is quite similar to the setup of ABA-fixed, but there are two differences: I used zeros instead of blank spaces, and I did not insert the additional tokens in the middle of the numbers. I do not exactly remember the result, but it was quite good, even though I still do not believe that this success can be interpreted as length generalization either. Have you tried not to put the blank spaces in between digits? What is good for the originally proposed method above this option? Can you provide some numerical results comparing them?
2. I personally love the idea of fine-tuning the model with ABA-var+RPE, starting from the baseline model (without any tricks like Position Coupling or Abacus), because this seems to be a remedy for the large training cost requirement I pointed out above. How long does it scale (in terms of the generalizable length)? What is the minimal amount of fine-tuning to achieve a similar performance as a model fully trained with ABA-var+RPE?

---

> ### Author Response · Authors · 2025-11-22
> **Response to Reviewer 9M9W (part I)**
>
> We thank the reviewer for the careful reading of our manuscript and for the insightful suggestions. These comments led us to add several new experiments and clarifications in the revised version, which we believe significantly strengthen the paper.
>
> ---
>
> >**Q1**: ABA-fixed is not length generalization.
>
> **A1**:  We agree with the reviewer that the training and test datasets share identical sequence lengths. However, the core of generalization lies in problem size, not strictly in sequence length. For arithmetic tasks, problem size corresponds to operand length. Although the sequence length is matched through blank spaces, the operands in the test set are significantly longer than those in the training set. Therefore, we argue that ABA-fixed still reflects generalization over larger problem sizes, even when the sequence length is matched.
>
> >**Q2**: The methods are not compared properly in terms of training cost.
>
> **A2**: We agree with the reviewer that this is a downside of ABA compared to position coupling and abacus embedding and we have made this point more explicit in section 7 (page 10). However, **the goal of this work is to improve generalization beyond the training length**, not to reduce training cost. Many prior methods (e.g., those involving scratchpads or long padding)  [Anil et al., Zhou et al., Shen et al.] also increase the sequence length and computational cost, yet their contribution lies in achieving better generalization, not efficiency.
>
> >**Q3**: Synergetic choice of training length and $p_{\max}$.
>
> **A3**: We thank the reviewer for the helpful suggestion. Following this recommendation, we trained ABA-fixed+RPE with $p_{\max}$=510 on addition up to 60 digits.The result is shown in Appendix E.3 (page32 Figure 23) in the revised Manuscript. Under this setting, the model generalizes reliably to 500-digit additions. This demonstrates that a larger ⁡$p_{\max}​$ can be used stably when the training sequence length is increased accordingly. In practice, ABA-fixed+RPE achieves roughly 90× length generalization for addition when the two hyperparameters are chosen in a compatible range.
>
> >**Q4**: Position coupling and Abacus embedding are not architectural change.
>
> **A4**:  We agree that if the Transformer implementation already supports custom position IDs, no additional code changes are required. However, Position Coupling and Abacus Embedding alter the core mechanism of positional representation by redefining how positions are encoded and combined with token embeddings. If we abstract away the architecture as a mathematical function $f_θ$, where θ denotes the learnable parameters, then positional coupling is essentially part of what the function f is. Changing the positional coupling changes the functional form $f_θ(x)$ of the model itself. Consequently, these methods are less flexible for finetuning or for tasks involving both text and arithmetic components, as the modified position logic can interfere with how the model processes natural language segments.
>
> >**Q5**: The success of ABA is because of short carry chains.
>
> **A5**: We appreciate the reviewer’s question. This is indeed a valid concern, and we did not consider this possibility carefully in the first version.
> We implemented the reviewer’s suggested parity task and its general m-parity variant.  In our experiments, ABA-fixed+RPE was trained on sequences of length 10 and evaluated on sequences of length 20. The result is in 4.4 (page8, figure6 right). The model reaches strong generalization on both mod-2 and mod-5 settings, confirming that ABA can handle tasks that require long-range propagation.
>
> To further examine the reviewer’s concern in the context of addition, we tested ABA-fixed on up to 20-digit addition with a full-carry test set, where the carry chain length equals the longer digit length (e.g., 9999999999 + 1). The result is in 4.4 (page8, figure6 left). Because our training data is sampled uniformly over digit lengths, full-carry cases rarely appear during training. Nevertheless, the model achieves perfect accuracy on the full-carry test set.
>
> This supports our hypothesis that the model learns a digit-wise rule for addition (and probably parity-scratchpad) that can be applied at any position seen in training, so performance does not degrade even when the carry chain spans the entire sequence. Together, these results suggest that the success of ABA is not limited to cases with short carry chains.
>
> >**Q6**: Have you tried not to put the blank spaces in between digits?
>
> **A6**: We believe the setup suggested by the reviewer closely resembles that in [Sabbaghi et al.], which we have included in Appendix D.1 (page 23, Figure 9). Empirically, ABA outperforms this variant. We think this is because inserting blank spaces between digits allows digits to appear at more diverse absolute positions during training, improving robustness to positional shifts at test time.

---

> > ### Author Response · Authors · 2025-11-22
> > **Response to Reviewer 9M9W (part II)**
> >
> > >**Q7**: More results on fine-tuning with ABA-var+RPE.
> >
> > **A7**:  Thank you for the helpful suggestion. We ran an additional experiment comparing the convergence behavior of models trained from scratch on up-to-10-digit addition with ABA-var and models first pretrained on the same addition dataset without ABA, then fine-tuned with ABA-var.  The new result is in Appendix D.2.2 (page24, figure 12), showing that ABA-var fine-tuned models converge much faster than models trained from scratch for $p_{\max}$ = 41 and 51, while $p_{\max}$ = 31 shows similar behavior for both training modes.
> > This confirms the reviewer’s hypothesis: **ABA-var fine-tuning is a compute-efficient alternative to training with ABA-var from scratch**, while reaching comparable final accuracy. We have added the new plot in the revised submission.
> > We again thank the reviewer for pointing out this useful advantage of the method.
> >
> > ---
> >
> > *Reference:*
> >
> > Anil, C., Wu, Y., Andreassen, A., Lewkowycz, A., Misra, V., Ramasesh, V., Slone, A., Gur-Ari, G., Dyer, E., & Neyshabur, B. (2022). Exploring length generalization in large language models. NeurIPS 2022.
> >
> > Sabbaghi, M., Pappas, G., Hassani, H., & Goel, S. (2024). Explicitly encoding structural symmetry is key to length generalization in arithmetic tasks. arXiv preprint, 2024
> >
> > Shen, R., Bubeck, S., Eldan, R., Lee, Y. T., Li, Y., & Zhang, Y. (2023). Positional description matters for transformers arithmetic. arXiv preprint, 2023.
> >
> > Zhou, H., Bradley, A., Littwin, E., Razin, N., Saremi, O., Susskind, J., Bengio, S., & Nakkiran, P. (2024). What algorithms can transformers learn? A study in length generalization. ICLR 2024
> >
> > **Final Note**: We sincerely appreciate the time and care you devoted to reviewing our work. We hope that the additional experiments and clarifications provided here satisfactorily resolve your concerns, and we would be grateful if you could consider a higher score in light of these revisions. If there are any remaining questions, we are happy to clarify them further.

---

> ### Comment · Reviewer_9M9W · 2025-11-27
>
> Thank you for your rebuttal which contains thoughtful responses and for the updates the manuscripts with additional experimental results. In particular, I am pleased to see the 500-digit extrapolation result with up to 60-digit training and using $p_{\max}=510$ (although I’m still curious about the result for the same experiment but with ABA-var). Still, I have some remaining questions.
>
> 1. **About length generalization and its categorization.**
>     - I agree that the essential size of the problems depends on the operand lengths. So if you situate ABA-fixed as a method only for ‘generalization in terms of operand lengths’, it might be a plausible claim.
>     - However, I don’t believe that the number of operand tokens(or, meaningful tokens) is the only factor hindering generalization capability of Transformers. I would say, from the viewpoint of “the only number of meaningful tokens matters”, a renowned synthetic task called NIAH (needle-in-a-haystack) has a small problem size, so it must be an easy task for Transformers…(but I don’t think it usually isn’t!) For this reason, I’m afraid I cannot agree with the opinion that the operand length is the only factor that determines the ‘problem size’.
>     - Under this spirit, I’d like to say that there are two different layers of length generalization: one is in the operand length; another is in the sequence length. With this **categorization** of length generalization problem, ABA-fixed is a method only for operand-length generalization, where as the other known methods (including Index Hinting, Abacus, Position Coupling, and ABA+var) are methods for both operand-/sequence-length generalization, despite their difference in length-generalizability. Obviously, achieving both is strictly more difficult than achieving one of them. Thus, I believe a strict comparison between ABA-fixed and the other methods is unfair.
> 2. **About the training cost.**
>     - Of course, I already know that the main contribution of this work is in improving the length generalization capability, instead of improving training efficiency. What I raised in my review was the fairness in comparison between the proposed method(s) and the baselines. Given that the training cost is identically restricted, can you still claim that ABA-fixed+RPE is the best, even when the methods proposed in previous works (Index Hinting, Abacus, Position Coupling) match or outperform its performance? For example, how will the test accuracy of ABA-fixed+RPE be if you only take the training cost used in the Abacus or Position Coupling papers?
>     - That said, I don’t think you can *never* claim the nice performance of ABA-fixed+RPE compared to the other methods. Even when it performs worse in a low-training-cost regime (e.g., using shorter training sequence lengths, less GPU time), you can still argue the significance of your method if it is strictly better than the others in a high-training-cost regime (e.g., matching the sequence length as test time, more GPU time). Nonetheless, it is possible only when you actually conduct more experiments comparing the methods given the identical training cost.
> 3. **About the m-parity task.**
>     - Thank you for conducting more experiments as per my suggestion! I’m mostly happy with the result. One minor concern is that, I would say, the test length is too short. What if you increase the test length to 200 (like in Figure 1)?
> 4. **About not putting blank spaces in the middle (Figure 9).**
>     - Just curious: have you put blank spaces of random lengths, both left and right sides of the operands? For example, if the operand length is 3 and the test length is 7, you should be able to train all of the following (although sampling them randomly):
>
>
>
>     \begin{align*}
>     \def\s{\square}
>     abc\s\s\s\s+def\s\s\s\s&=ghij\s\s\s\s \\\\
>     \s abc\s\s\s+\s def\s\s\s&=\s ghij\s\s\s \\\\
>     \s\s abc\s\s+\s\s def\s\s&=\s\s ghij\s\s \\\\
>     \s\s\s abc\s+\s\s\s def\s&=\s\s\s ghij\s \\\\
>     \s\s\s\s abc+\s\s\s\s def&=\s\s\s\s ghij
>     \end{align*}
>
>
>     - As far as I can see in Appendix D.1, it seems like it only displays the first example above.
>
> Again, thank you for the time you spent writing the detailed rebuttal. For now, however, I retain my score as 4.

---

> ### Author Response · Authors · 2025-12-03
> **Response to Reviewer 9M9W**
>
> We thank the reviewer for the detailed and thoughtful response. Below, we answer the questions the reviewer raised:
>
> > **Q1**: ABA handles only one of two categories of length generalization, while existing methods handle both.
>
> **A1**: We thank the reviewer for raising the important point that length generalization can be categorized into two types: sequence-length and operand-length generalization.
>
> At the same time, we respectfully disagree with the statement that “ABA-fixed is a method only for operand-length generalization, whereas the other known methods are methods for both operand-/sequence-length generalization”. For most effective methods proposed for arithmetic length generalization, such as Index Hinting, Position Coupling, the extra position markers are placed to digits of the same significance. As a result, we view them as methods which primarily aim to improve generalization over operand length; sequence length generalization then follows as they either do not add extra tokens or sequence length grows in proportion to operand length (i.e. a1b2c3+a1b1c1=a3b3c4). In contrast, for tasks where the input is much less structured, such as the NIAH task the reviewer mentioned, these methods do not directly help with sequence-length generalization. As such, these methods **do not reflect an extra ability for sequence length generalization when compared to ABA**.
>
> To make this clear, we have clarified this distinction between sequence-length and operand-length generalization in Section7 (page10) in the revised version, and we now state explicitly that ABA-fixed targets the latter, while other baselines also handle sequence-length generalization.
>
> >**Q2**: Is ABA fairly evaluated against baselines, especially when the training cost is fixed?
>
> **A2**:  We thank the reviewer for raising this important point. In response, to fairly compare ABA with baselines while keeping training cost fixed, we ran experiments comparing the evolution of validation accuracy as a function of wall-clock training time for ABA and the baseline methods. Although the underlying model configurations are not identical, Appendix C.4 (Page22, figure9) show that ABA is indeed significantly slower than Position Coupling under comparable settings. We have made this limitation explicit in the revised manuscript, and we now emphasize that ABA mainly offers an advantage in a high-training-cost regime, where a larger budget in tokens and time is acceptable.
>
> >**Q3**: What if you increase the test length to 200 in the m-parity task?
>
> **A3**:  We thank the reviewer for their compliments on our empirical results on the m-parity task. To address the reviewer’s request for a test length of 200, we have done an experiment on the m-parity task when trained on a 50-digit parity dataset, and $p_{\max}$=201. Thus, the test length is 200. As shown in Appendix D.3.2 (page 29, Figure20), ABA-fixed+RPE reaches near-perfect accuracy.
>
> >**Q4**: How does ABA compare to putting blank spaces of random lengths on both the left and right sides of the operands?
>
> **A4**: We thank the reviewer for raising this alternative baseline. We have conducted an experiment using the format the reviewer proposed. As shown in Appendix D.1 (page 23, Figure10), it reaches marginal length generalization compared to ABA.

---

### Official Review · Reviewer_SqC2 · 2025-11-03

**Soundness:** 3
**Presentation:** 4
**Contribution:** 2
**Rating:** 6
**Confidence:** 4

**Summary:**

The paper studies the problem of length generalization for arithmetics (addition and multiplication) with transformers. Several works have proposed solutions for this problem via changing the architecture (e.g., positional embeddings) or padding all of the inputs. This work claims that they can preserve the accuracy once they surpass the training lengths by none of the previous changes. Instead, they propose a novel augmentation method.

**Strengths:**

1- The paper is well-written and easy to understand. They have also included extra explanations and comparisons in the appendix that makes the paper complete.

2- As far as I am aware, adding zeros for augmentation is investigated in previous work, but randomly adding blank tokens is novel.

3- The experiments clearly show the benefits of using the paper's proposed method. They also cover most of the existing baselines in their comparisons that have made it more convincing.

4- The results for ABA + scratch-padding in multiplication look promising considering that there are no architectural changes.

5- The paper adapts the results of [Huang et al.] to explain why their method can address the length generalization problem in theory.

**Weaknesses:**

1- I am not certain about the distinction of augmentation with zeros that has been investigated before (e.g, multiplying both operands by 10^n) and the proposed solution that randomly adds blank spaces between digits (as opposed to only adding them to the two sides of the operands). I also did not find any arguments about this in the paper and I think this needs to be added. Even though the theory section justifies their results with the result of RASP, I think authors need to motivate this more intuitionally.

2- The paper motivates their results by claiming for no architectural changes. However, as Table 1 shows, the performance indeed depends on the type of the positional encoding and this seems to be an important factor.

3- The comparisons in the multiplication section is quite limited in the main body, and something similar to Figure 13 has to be added there.

**Questions:**

Pls see above for questions.

---

> ### Author Response · Authors · 2025-11-22
> **Response to Reviewer SqC2**
>
> We thank the reviewer for the encouraging feedback and detailed assessment of our work, as well as for the constructive suggestions that helped us improve the paper.
>
> ---
>
> >**Q1**:  How does the usage of zero padding and random blank spaces relate to prior work?
>
> **A1**: Thank you for pointing out this missing connection. We have clarified in related work (page2) that our method builds on prior zero-padding format. Zeropadding is first introduced in [Lee. et al], and has been used in several papers in arithmetic length generalization. In those settings, zeros are added only at the left or right of each operand and result, so that corresponding digits always occupy fixed absolute positions once padded. In our work, we use zero-padding for the same purpose of aligning digits.
>
> Randomly blankspaces insertion is first introduced in [Shen et al]. We explicitly compare their random-spacing scheme to our ABA in Appx. D.2.2 and find that alignment of the inserted positions is crucial for strong length generalization.
>
>
> >**Q2**:  The claim of “no architectural changes” may be at odds with the dependence on positional encodings.
>
> **A2**: The phrase “no architectural changes” in our paper refers to not modifying the Transformer code or the attention equations: ABA is a data augmentation that can be combined with many existing positional encoding schemes.
> Table 1 exhibits that ABA improves length generalization for a broad range of positional encodings. It is expected that the performance will vary across positional embeddings, as length generalization is very sensitive to positional information [Kazemnejad et al]. Other methods such as index hint [Zhou et al] and Abacus embedding [Mcleish et al] also show to have different performance with different positional embeddings.
>
> >**Q3**: A per-digit-length test result for multiplication should be included.
>
> **A3**: Thank you for this helpful suggestion. We have followed your recommendation and added a per-digit evaluation for multiplication in Appendix D.3.4 (page 29, Fig.20). In addition, the current version already includes a 2D heatmap for multiplication (Fig.6), where the horizontal and vertical axes correspond to the digit lengths of the first and second operands. We view this heatmap as complementary to the per-digit curves, since it shows accuracy for every pair of operand lengths at once and makes the asymmetry between the two operands more transparent.
>
> ---
>
> **Final Note**: Thank you again for your detailed and thoughtful comments. If our responses have addressed your concerns, we would be very grateful if you would consider updating your score and supporting the paper’s acceptance. Please let us know if any issues remain unclear. We appreciate your time and careful evaluation.
>
>
> *Reference:*
>
> Lee, Nayoung; Sreenivasan, Kartik; Lee, Jason D.; Lee, Kangwook; Papailiopoulos, Dimitris (2024). Teaching Arithmetic to Small Transformers. ICLR 2024.
>
> Kazemnejad, A., Padhi, I., Natesan Ramamurthy, K., Das, P., & Reddy, S. (2023). The Impact of Positional Encoding on Length Generalization in Transformers. NeurIPS 2023.
>
> McLeish, S., Bansal, A., Stein, A., Jain, N., Kirchenbauer, J., Bartoldson, B. R., Kailkhura, B., Bhatele, A., Geiping, J., Schwarzschild, A., & Goldstein, T. (2024). Transformers Can Do Arithmetic with the Right Embeddings. NeurIPS 2024.
> Zhou, H., Bradley, A., Littwin, E., Razin, N., Saremi, O., Susskind, J., Bengio, S., & Nakkiran, P. (2024). What algorithms can transformers learn? A study in length generalization. ICLR 2024.
>
> Shen, R., Bubeck, S., Eldan, R., Lee, Y. T., Li, Y., & Zhang, Y. (2023). Positional description matters for transformers arithmetic. arXiv preprint, 2023.
>
> Zhou, H., Bradley, A., Littwin, E., Razin, N., Saremi, O., Susskind, J., Bengio, S., & Nakkiran, P. (2024). What algorithms can transformers learn? A study in length generalization. ICLR 2024.

---

### Author Response · Authors · 2025-12-03
**Summary**

Dear Area Chair,

Thank you for handling our submission. In view of the OpenReview incident, we briefly summarize the reviews and our rebuttal.

Reviewer SqC2 was overall positive about our result. They asked how ABA relates to prior zero padding and random spacing, how “no architectural changes” fits with positional encodings, and requested per-digit multiplication results. We clarified these points in the page2 related work and added the requested experiments.

Reviewer 9M9W appreciated the experiments and theory but questioned whether ABA-fixed should count as length generalization, whether training cost comparisons are fair, and whether ABA scales to very long carry chains. In response, we added experiments on 500-digit addition after training up to 60 digits, full-carry addition, as well as mod-2 and mod-5 parity. In their follow-up, the reviewer complimented our additional experiments but still had questions  about length generalization definitions and training cost comparison. In response, we clarified the distinction between operand-length and sequence-length generalization and added cost-matched training curves.

Reviewer F1EX found the method clear and effective but was concerned that ABA trains on long sequences, whether multi-operand addition truly generalizes in high-carry settings, and whether it’s fair for other baseline in our Generalization from 10+10 experiment. We added stress tests for high-carry multi-operand addition, full-carry addition, and a control experiment where Position Coupling is evaluated with matched sequence length.

Reviewer 2GoR praised our method as simple yet effective but asked why not simply train on longer operands; how exactly the sorting task and placeholders work; and questioned the tightness of our communication complexity bound for random blank space augmentation in the $K \in \Theta(\log p_{\max})$ case.  To clarify the first question, we demonstrated that prior work shows direct training on longer operands does not by itself fix length generalization. To address the second question, we described the sorting setup in detail. To address the third question, we clarified that our provided lower bound was tight for the $K = \Theta(L)$ case, but also provided a lower bound for the $K \in \Theta(\log p_{\max})$ case which matched the upper bound the reviewer suggested. We have updated Section 6 and Appx. H with these results.

We again thank the reviewers’ detailed and helpful suggestions and regret that there was limited opportunity for further exchange. We are confident that our responses solve their questions and kindly ask the AC to take a close look at the rebuttals.

Thank you,


The Authors

---

### Meta-Review · Area_Chair_CXxT · 2026-01-05

**Summary:**

## Summary
This submission proposes Aligned Blankspace Augmentation (ABA) for (operand-)length generalization of Transformers on algorithmic tasks, with a primary focus on addition. ABA amounts to inserting jointly aligned blank spaces to all operands and answers in the addition problems. Without altering the transformer architecture (such as using a specialized position embedding), ABA achieves superior operand-length generalization, outperforming other baselines. Other algorithmic tasks such as copy, reverse, multiple addition, sorting, and $N \times 2$-digit multiplication are tested, and ABA shows good performance on some of the methods here. The authors also try ABA with scratchpads to solve $M \times N$-digit multiplication and show that scratchpad helps for multiplication, as also evidenced by a prior work (Cho et al 2025).

## Reviewer Concerns
Major concerns raised by reviewers can be summarized as the following.
- **”Is this really length-generalization?”/Increased length of training sequences**. Reviewers 9M9W and F1EX expressed concerns if the proposed method can truly be viewed as a length generalization method, as the model is exposed to lengthy training sequences that match the test length. A related question raised by Reviewers F1EX and 2GoR is: why not just construct training sets with longer operands, if the $p_{max}$ is capped anyway? Reviewer SqC2 also asked for a clarification of the difference from “augmentation with zeros”.
- **Training Cost**. Reviewer 9M9W pointed out that the proposed method has larger training cost compared to baselines, stemming from the increased sequence length.
- **Claims on “no architectural changes”**. The paper claims that ABA achieves length generalization without architectural changes, but is still dependent on the choice of positional encoding (Reviewer SqC2). Reviewer 9M9W claimed that Abacus and Position Coupling can also be viewed as “modification-free”.
- **Carry propagation test**. Reviewers 9M9W and F1EX asked if additional results demonstrating long-range carry propagation can be added. Reviewer F1EX also asked if the large carries in multi-operation addition task are really handled by the model properly.
- **Comparisons to other baselines**. Reviewer SqC2 pointed out that comparisons in the multiplication section is limited, Reviewer F1EX questioned if the comparisons in the “generalization from 10+10” experiment is fair, and Reviewer 2GoR noted that the performance in Table 2 does not show consistent improvements over position coupling.
- **Theory**. Reviewer 2GoR had some concern about the correct communication complexity in the theory section.

**Reviewer Concerns:**

Unfortunately, the discussion period closed before we received feedback from three out of the four reviewers. Based on my reading, the authors’ responses can be summarized as
- **”Is this really length-generalization?”/Increased length of training sequences**.
  - The authors clarified that although sequence lengths are the same, operand lengths differ between train and test data, which they view as the “problem size.” Reviewer 9M9W responded to the rebuttal, categorizing “operand-length generalization” (this is where ABA belongs to) vs “sequence-length generalization”, and the authors’ follow-up revision also reflects this.
  - As for the question “why not train on longer sequences?”, the authors clarified that generalization beyond the training distribution is the key here. While I understand the distinction the authors are pointing to, I still wonder: at least for ABA-fixed (the version with superior performance), isn’t the test length capped by $p_{max}$ anyway, and we train on length-$p_{max}$ sequences? If the end goal is to do well on target length $p_{max}$, then why not take the reviewers’ suggested route? **I think these are valid criticisms to the proposed approach**.
  - Question by Reviewer SqC2 was addressed properly.

- **Training Cost**. The authors admitted the downside and added a note on this in the paper. They emphasize that the primary goal of ABA is length generalization, not efficiency. Reviewer 9M9W followed up saying that the comparisons should be fair only if the training compute budgets match. The authors were upfront about this and added a plot (Figure 9) showing that ABA is indeed slower (I appreciate the honesty). High training cost could be a major limitation because **the primary motivation for studying length generalization is the efficiency of transformers in long contexts.**

- **Claims on “no architectural changes”**. The authors clarified that “no architectural changes” refers to not modifying the Transformer code or the attention equations. In the response to Reviewer 9M9W, the authors pointed out that although position coupling and abacus do not require code-level changes, they alter conventional position embedding to a special version tailored for the specific task at hand. I think the concerns here are well-addressed.

- **Carry propagation ablation**. The authors added Section 4.4 to handle these questions. ABA shows good performance on Full-carry addition and parity. However, for multi-operand addition, the length generalization in terms of the number of operands **does deteriorate**. The plots show training on five operands and testing on ten operands, but I expect a worse breakdown when the number of operands becomes even larger (e.g., when there are 12 operands, the carry extend to an extra digit: 9*12 = 108).

- **Comparisons to other baselines**. The authors added plots on the multiplication results and re-ran the 10+10 experiment for position coupling with fairer setups but observed that it still falls behind ABA. For Reviewer 2GoR’s question on Table 2, the authors admitted that ABA does not dominate on all tasks and emphasized ABA’s values. **However, the appeal to the “no free lunch theorem” appears misplaced in this context**.

- **Theory**. The authors clarified the range of $K$ considered in the setting and added a new lower bound.

**Reviewer Scores:**

The initial reviews had scores 6/4/4/4. Here is what I expect about the reviewers’ final assessment, focusing on the negative reviewers:
- **Reviewer 9M9W** (score 4) engaged in the discussion but did not feel inclined to raise their score at the moment. Given the reviewer’s concern on the training cost, I expect that the reviewer’s evaluation still remains the same.
- The main concern of **Reviewer F1EX** (score 4) was whether the proposed method could be really called “length-generalization” when it requires exposing the model to long sequences during training. As I wrote above, while I see the authors’ point, the criticism still remains outstanding. Considering the nontrivial-but-not-impressive performance on high-operand-count experiments as well, I believe that Reviewer F1EX wouldn’t have raised their score.
- For **Reviewer 2GoR** (score 4), I think the authors’ response does not fully address Q1 and Q2. Additional follow-up comments could have resolved the issues, but I think the expected score for now is 4.

The operand-length generalization demonstrated by ABA-fixed is impressive, and it is particularly valuable because the method does not require changing architectural components such as positional encoding. However, this comes at the cost of requiring training length to match the test length, and it’s missing something about the core motivation of length generalization: isn’t the whole point about “train on shorter, generalize to longer”? Having to train the model on the full-length sequence causes massive training cost, at the same time bringing up the question of “if the max length is fixed anyway, why not just train on longer digit examples?”. Unfortunately, I think the paper sits slightly below the bar for ICLR.

---

### Decision · Program_Chairs · 2026-01-26

Reject